# Accommodating individual travel history and unsampled diversity in Bayesian phylogeographic inference of SARS-CoV-2

Philippe Lemey [1✉], Samuel L. Hong [1], Verity Hill [2], Guy Baele[1], Chiara Poletto[3], Vittoria Colizza [3], Áine O'Toole [2], John T. McCrone[2], Kristian G. Andersen[4], Michael Worobey[5], Martha I. Nelson[6], Andrew Rambaut [2] & Marc A. Suchard [7,8,9✉]

Spatiotemporal bias in genome sampling can severely confound discrete trait phylogeographic inference. This has impeded our ability to accurately track the spread of SARS-CoV-2, the virus responsible for the COVID-19 pandemic, despite the availability of unprecedented numbers of SARS-CoV-2 genomes. Here, we present an approach to integrate individual travel history data in Bayesian phylogeographic inference and apply it to the early spread of SARS-CoV-2. We demonstrate that including travel history data yields i) more realistic hypotheses of virus spread and ii) higher posterior predictive accuracy compared to including only sampling location. We further explore methods to ameliorate the impact of sampling bias by augmenting the phylogeographic analysis with lineages from undersampled locations. Our reconstructions reinforce specific transmission hypotheses suggested by the inclusion of travel history data, but also suggest alternative routes of virus migration that are plausible within the epidemiological context but are not apparent with current sampling efforts.

[1] Department of Microbiology, Immunology and Transplantation, Rega Institute, KU Leuven, Laboratory of Clinical and Evolutionary Virology, Leuven, Belgium. [2] Institute of Evolutionary Biology, University of Edinburgh, Edinburgh EH9 3FL, UK. [3] INSERM, Sorbonne Université, Institut Pierre Louis d'Epidémiologie et de Santé Publique IPLESP, F75012 Paris, France. [4] Department of Immunology and Microbiology, Scripps Research, La Jolla, CA 92037, USA. [5] Department of Ecology and Evolutionary Biology, University of Arizona, Tucson, AZ 85721, USA. [6] Division of International Epidemiology and Population Studies, Fogarty International Center, National Institutes of Health, Bethesda, MD 20892, USA. [7] Department of Biomathematics, David Geffen School of Medicine, University of California Los Angeles, Los Angeles, CA 90095, USA. [8] Department of Biostatistics, Fielding School of Public Health, University of California Los Angeles, Los Angeles, CA 90095, USA. [9] Department of Human Genetics, David Geffen School of Medicine, University of California Los Angeles, Los Angeles, CA 90095, USA. ✉email: philippe.lemey@kuleuven.be; msuchard@ucla.edu

Since its emergence in late 2019, SARS-CoV-2 has rapidly spread across the world, prompting governments to enact restrictions on human mobility that are unprecedented on a global scale. While coronavirus disease 2019 (COVID-19) has exposed critical gaps in public health preparedness and research, an observed strength of the COVID-19 response has been the rapid speed with which whole-genome sequences of SARS-CoV-2 have been generated globally (over 100,000 genomes as of September 15, 2020). The success of global sequence production has at least been partially facilitated by protocols and networks that arose during the response to the 2013–2016 Ebola virus disease (Ebola) epidemic in West Africa. That Ebola epidemic was particularly important for spurring development of tools and methods for real-time in-country virus sequencing, including in resource-limited settings, e.g., ref. [1].

Genomic data represent a key resource for testing hypotheses about how and when SARS-CoV-2 became established in different locations. For example, phylogenetic approaches may be able to distinguish community transmission from new introductions from travelers[2], whether viral outbreaks were associated with multiple introductions[3], how long viruses may have been transmitting undetected in a community ("cryptic transmission"[4]) and when widespread domestic spread began in the United States[5]. Such studies have already greatly contributed to charting the course of an unfolding pandemic, informing public health decisions about when to enforce lockdown measures of various degrees of stringency. Recently, many countries have entered a new phase in which restrictions, such as school and business closures, and mobility restrictions are eased or lifted. Suppressing SARS-CoV-2 spread remains our only viable defense so far and efficient test, trace and isolate systems will be crucial tools to achieve these goals. Molecular epidemiology can continue to inform public health actions during this phase, for instance through uncovering cryptic transmission or new introductions, as sources of flare-ups.

Recent advances in virus sequencing and phylogenetics hold great promise for addressing key questions in infectious disease epidemiology and outbreak response[6]. There are limitations, however, to the insights that can be obtained from the wealth of SARS-CoV-2 genome data. The current sequence diversity is relatively limited because SARS-CoV-2 emerged only recently in late 2019 and because SARS-CoV-2 transmission outpaces the rate at which substitutions accumulate[4]. This implies that short-term transmission patterns may not leave a detectable footprint in virus genomes, resulting in poorly resolved genomic reconstructions. In addition, large spatiotemporal biases exist in the available genome data, e.g., [7]. For instance, ~40% of currently available genomes have been sampled from the UK, whereas Italy, having experienced a similar number of cases and likely an earlier epidemic onset, only represents ~0.3% of the genome collection on GISAID[8]. Both low sequence diversity and sampling bias confound the interpretation of transmission patterns, and highly similar SARS-CoV-2 genomes from the same or different locations do not necessarily imply direct linkage.

Despite a relatively slow evolutionary rate, the "phylodynamic threshold" for SARS-CoV-2 was reached relatively early[9], meaning that sufficient divergence had accumulated over the sampling time range to infer time-calibrated phylogenies and the underlying transmission processes that generate such trees, including spatiotemporal spread. However, sampling bias presents a critical challenge for popular discrete trait ancestral reconstruction procedures[10,11]. Although the modern phylodynamic framework includes other statistical approaches that are less sensitive to sampling bias, e.g., [12–14], computational complexity challenges their application to large data sets, in particular when insights are needed in short turnaround times.

This explains the widespread adoption of ancestral reconstruction approaches that provide real-time tracking of pathogen evolution and spread[15].

When confronted with low diversity and sampling bias, evolutionary reconstructions may greatly benefit from integrating additional sources of information. Bayesian phylodynamic approaches are particularly adept for this purpose[16], and phylogeographic methods in particular have been extended to take advantage of human transportation data as proxies of population-level connectivity between locations[17]. This approach has been utilized in a wide range of applications, including the identification of the key drivers of *Ebolavirus* spread in West Africa[18]. Individual travel history of sampled patients also represents an important source of information that currently has not been used to its full potential in phylogeographic inference. Genomic data from (returning) travelers may help to uncover pathogen diversity in locations that are otherwise undersampled. This has been elegantly demonstrated by a study on Zika virus that used travel surveillance and genomics to demonstrate hidden viral transmission in Cuba[19]. Formally integrating such travel data in phylogeographic reconstructions may therefore help to address or correct for sampling bias. In general, epidemiological information provides important context to assess genomic sampling biases, and this can be used to subsample genomes by location in situations where large collections are available[20]. The question therefore emerges how such information can be formally embedded in phylodynamic models. Specifically, if particular locations remain undersampled despite their potential importance for viral spread, can the reconstructions account for hypotheses alternative to the ones supported by the sampled genomes, but plausible according to the epidemiological context? Here, we extend phylogeographic methodology to incorporate travel data and, together with the integration of transportation data, we apply this to reconstruct the early spread of SARS-CoV-2 and formally validate the approach, using a posterior predictive accuracy procedure. In addition, we demonstrate how epidemiological data can be used to incorporate unsampled diversity. Taken together, these approaches constitute an important step toward more realistic and more nuanced phylogeographic reconstructions.

## Results

**Travel history uncovers more realistic phylogeographic patterns**. To focus on the early dynamics of SARS-CoV-2 spread, we analyze a data set consisting of curated genomes available in GISAID on March 10th, 2020 ($n = 282$). Having collected travel history data for over 20% of the sampled patients, we extend phylogeographic reconstruction methodology to incorporate this source of information (cfr. "Methods"). Specifically, we augment the sampled genomes from known travelers with their recent travel location and either the time of their return journey or estimated time of infection (see below). Critically, we include travelers returning from severely undersampled locations, including Italy, Iran, and Hubei, China, where the virus was first identified.

In our Bayesian analysis of the complete data set, we model a discrete diffusion process between 44 locations: 29 countries and 15 locations within China, including 13 provinces, one municipality (Beijing), and one special administrative area (Hong Kong). We fit a generalized linear model (GLM) parameterization of the discrete diffusion process and consider air travel data, within-continent geographic distances, and an estimable asymmetry coefficient for transitions from and to Hubei as covariates for the diffusion rates. In Table 1, we compare the posterior estimates for the inclusion probabilities and conditional effect sizes (on a log

**Table 1 Inferred generalized linear model (GLM) of discrete location transitions under three different usage strategies of travel history information.**

| Information usage strategy | Air travel | | Geographic distance | | Asymmetry out of Hubei | |
|---|---|---|---|---|---|---|
| | Inclusion probability | Log conditional effect size | Inclusion probability | Log conditional effect size | Inclusion probability | Log conditional effect size |
| Sampling location and travel history | >0.999 | 0.718 (0.505, 0.948) | 0.114 | −0.042 (−0.105, 0.024) | >0.999 | 2.362 (1.864, 2.859) |
| Sampling location only | >0.999 | 1.131 (0.838, 1.481) | 0.049 | −0.016 (−0.105, 0.077) | 0.969 | 1.620 (0.631, 2.319) |
| Travel origin location | >0.999 | 1.000 (0.709, 1.317) | 0.114 | 0.050 (−0.043, 0.146) | >0.999 | 2.226 (1.650, 2.800) |

We report posterior inclusion probabilities and posterior mean (95% highest probability density intervals) log conditional effect sizes for air travel, geographic distance, and asymmetry out of Hubei.

scale) of these covariates in an analysis that incorporates (i) sampling location and travel history (travel-aware phylogeographic inference), (ii) sampling location only, and (iii) travel origin location. Regardless of what location data we use in the analyses, they consistently indicate that in this early stage, SARS-CoV-2 spread is shaped by air travel and not by geographic distance, and that there is strong asymmetric flow out of Hubei. Interestingly, this asymmetry is somewhat stronger for the analyses that incorporate travel locations compared to the analysis using only sampling locations. One explanation is that the majority of travelers were returning from Hubei, and adding this information contributes appropriately to the intensity of outflux from Hubei. In Supplementary Text S3, we also report estimates for the same predictor set expanded with sample sizes as origin and destination predictors for transitions. These estimates indicate that sample sizes contribute considerably less in travel-aware analyses compared to using sampling location only, suggesting that the travel-aware reconstructions will also be more robust to sample size bias.

As expected for a low degree of sequence variability over this limited time range, our Bayesian phylogeographic reconstructions are burdened by a high degree of topological uncertainty. In Fig. 1, posterior support for the nodes in the maximum clade credibility (MCC) tree is represented by the size of the node circles and the support across all nodes is summarized as a histogram. This illustrates that only a limited number of clusters are reasonably well supported. Poor phylogenetic signal is also illustrated by likelihood mapping analysis (Supplementary Fig. S4). This renders a single phylogenetic tree summary inadequate to interpret phylogeographic history. We sidestep this problem by studying spatial trajectories that marginalize over phylogenetic variability, in the ancestral history of single taxa.

In Fig. 2, we consider a case study involving the spatial path of a virus that was collected in Switzerland in February 2020 (EPI_ISL_413021). As summarized in Fig. 2a, the Swiss virus is positioned within a cluster of viruses primarily from Europe that has been the subject of controversy. The basal clustering of a genome from the first detected case in Germany led to the speculation that the virus spread from Germany to Italy[21]. The trajectory plots summarize across the posterior distribution the time intervals in the phylogenetic ancestry during which the virus remains in the same location (horizontal lines) and the transitions between two locations (vertical lines). Using the standard location of sampling (Fig. 2b), the trajectory plot traces the origins of the Swiss virus back to the Netherlands, an inference that is likely due to a relatively large number of Dutch genomes in the cluster. Going further back in time, the virus appears to have spread from Germany to the Netherlands. The original ancestry prior to Germany becomes uncertain, and could be Hubei, Guangdong, or

other locations. Italy is not part of the trajectory at all because Italy is undersampled (only two genomes in the cluster). Using the locations of origin for the travelers (Fig. 2c), the trajectory is more ambiguous about the spatial path, and whether the Swiss virus came from Italy or the Netherlands. The travel-aware reconstruction, which includes both sampling location and traveler's location of origin (Fig. 2d), almost fully resolves the ancestry of the Swiss virus. The Swiss virus was imported from Italy, and not the Netherlands. The fact that this cluster contains five genomes from travelers returning from Italy to various countries, including Germany, Scotland, Mexico, Nigeria, and Brazil, is instrumental in positioning Italy at the root of this cluster and helps correct for Italy's lack of data. The inclusion of genomes associated with a Hubei travel history also strengthens the original Hubei ancestry in the phylogeography, as trajectories appear to coalesce earlier in Hubei. Although the trajectory suggests an introduction into Italy from Germany, the support for this is not overwhelming and solely due to the single genome sample from Germany basal to the Italian cluster. We return to this transmission hypothesis in the next section.

In a second case study, we consider a virus from Australia sampled in February 2020 (EPI_ISL_412975). The virus is positioned within a clade of other closely related viruses from Australia (lineage B.4, Fig. 1), some of which were sampled from travelers returning from Iran[22]. Using sampling location alone does not provide any support for Iranian ancestry, since the data set does not include any genomes directly sampled from Iran (Fig. 3a). Using the locations of origin for the travelers does support Iranian ancestry (Fig. 3b), but with considerable ambiguity. However, the travel-aware reconstruction, including both sampling location and traveler's location of origin, clearly supports an ancestry that includes Iran (Fig. 3c). This Iran–Australia case study provides an example where enforcing the travel location somewhat deeper in the evolutionary history (at return dates of the travelers) imparts more information that is critical for correctly reconstructing ancestral relationships.

Although our case studies illustrate the impact of incorporating travel history information and how this results in more realistic hypotheses, they do not provide a formal assessment of the approach. To validate the approach, we perform a posterior predictive accuracy assessment. Specifically, we perform a tenfold cross validation that, in each fold, holds out 10% of the travel history information in the phylogeographic inference (cfr. "Methods"). We estimate the ancestral travel locations for the tips with withheld travel information and compare the prediction accuracy (i) when including the remaining 90% of the travel history and (ii) when excluding all travel history data, using the Brier score (BS; cfr. "Methods"). Our evaluation results in a BS of 1.35 without travel history data, and 0.70 with 90% of the travel

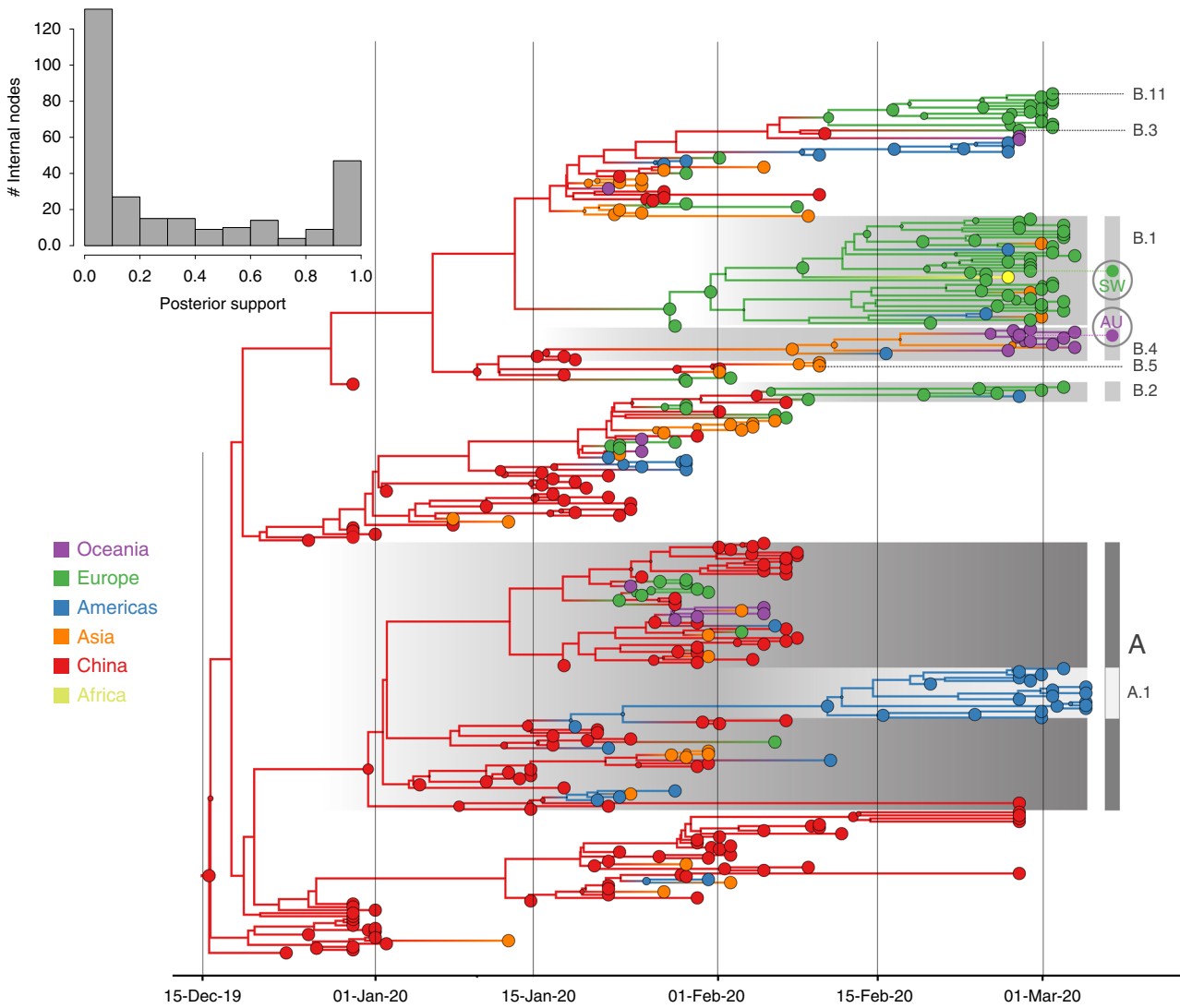

**Fig. 1 Bayesian phylogeographic reconstruction for the full data set incorporating travel history data.** Although the phylogeographic analysis was performed using 44 location states, nodes and branches are shaded according to an aggregated color scheme for clarity. Lineage classifications are highlighted for specific clusters: lineage A is embedded in lineage B. For lineage B, only specific sub-lineages are indicated. The taxa from Switzerland (SW) and Australia (AU) further investigated using trajectory plots are indicated at the tips of the trees. The inset represents a histogram of the internal node posterior support values.

history data. This represents a large, twofold improvement in discrimination and calibration. In practical terms, the odds of identifying the correct ancestral state increases 4.5-fold, which clearly demonstrates the advantages of including travel history under conditions where the truth is known.

**Unsampled diversity reinforces reconstructions informed by travel data and unveils alternative transmission hypotheses.** To further explore the sensitivity of phylogeographic analyses to sampling bias, we incorporate unsampled taxa in our reconstructions, in addition to travel history data. Supplementary Text S2 illustrates how this approach can recover established pathways of migration using a Zika virus example. In our SARS-CoV-2 analyses, we add unsampled taxa for locations that are undersampled according to case counts (cfr. "Methods"), in this case primarily for Hubei ($n = 307$), followed by Italy ($n = 47$), Iran ($n = 40$), and South Korea ($n = 30$). We specify a prior distribution over their tip ages ("sampling times") based on estimates of prevalent infections (cfr. "Methods"). Using this

framework, we revisit the trajectory estimate for the Swiss B.1 genome (Fig. 4). In contrast to the reconstructions with no unsampled taxa (Fig. 2), we now mainly observe a direct transition from Hubei to Italy (posterior probability = 0.88), implying that a second introduction from Hubei that is independent from the introduction into Germany may have seeded the Italian clade. This hypothesis arises from the inclusion of unsampled Hubei taxa that now cluster between the German virus and the Italian clade (Fig. 4), even though the branch connecting the German genome to the Italian clade only represents a single substitution. The many unsampled Italian taxa that fall in this clade further reinforce its Italian ancestry.

The inclusion of unsampled taxa also provides more resolution on the Hubei ancestry of the Iran–Australia case study in lineage B.4 (EPI_ISL_412975). We estimate a similar trajectory as the analysis using travel history without unsampled taxa (Fig. 3), but with a more recent coalescence in Hubei because of unsampled Hubei taxa clustering basal to an Iranian clade (Fig. 5). The genomes from four Australian travelers returning from this

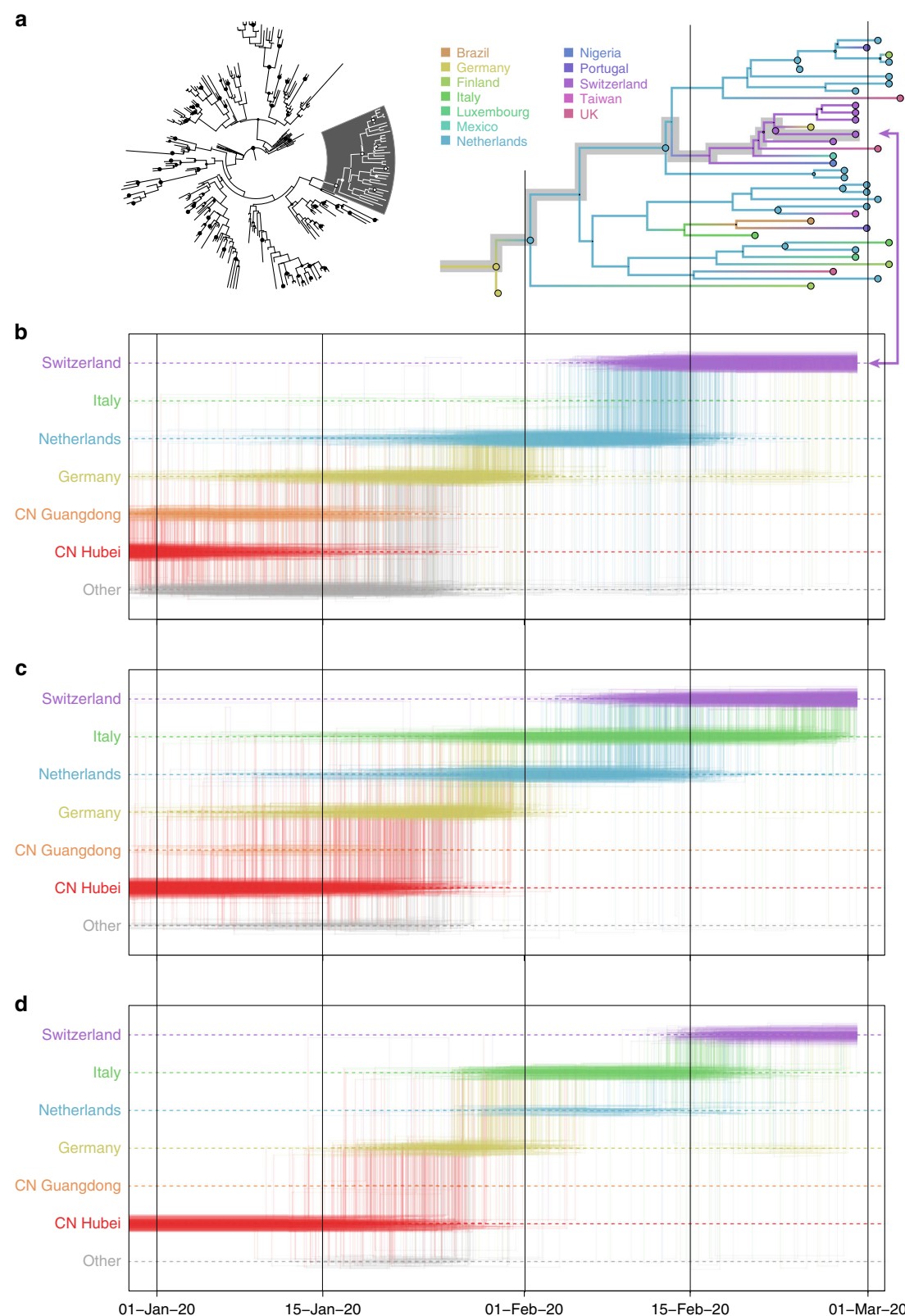

country, one direct contact, another Australian genome without travel history, as well as a genome from a traveler returning from Iran to New Zealand are effectively embedded in unsampled Iranian diversity. The most basal virus in this clade is from a Canadian traveler returning from Iran. Although the basal nature of this virus is not well supported, there is good posterior support for the monophyly of all the sampled genomes in this clade.

Notably, this virus was sampled before the first report of COVID-19 in Iran on February 19th[23], but our reconstruction suggests that considerable diversification, and hence transmission, already took place prior to this report.

In addition to focusing on specific transmission patterns, we also summarize the overall dispersal dynamics in a way that marginalizes over plausible phylogenetic histories for the different

**Fig. 2 Phylogeographic reconstruction and spatiotemporal ancestry of a virus collected in Switzerland (EPI_ISL_413021). a** Phylogenetic cluster with the Swiss virus shaded in gray in the MCC tree, and the same B.1 cluster with branches colored according to posterior modal location states inferred by an analysis using sampling location only. The tip for the Swiss virus corresponding to the trajectory in **b** is indicated with an arrow. Markov jump trajectory plot depicting the ancestral transition history between locations from Hubei up the sampling location for the Swiss genome, using **b** sampling location only, **c** travel origin location, and **d** sampling location and travel history. The trajectories are summarized from a posterior tree distribution with Markov jump history annotation. A horizontal line in a trajectory represents the time during which a particular location state is maintained in the spatiotemporal ancestry of the virus. An example of such an ancestry is highlighted in gray in the MCC tree cluster. A vertical line represents a Markov jump between two locations in the trajectory. The most prominent locations in the posterior trajectories are ordered along the Y-axis together with "other", which represents all remaining locations. The relative density of lines reflects the posterior uncertainty in location state and transition time between states.

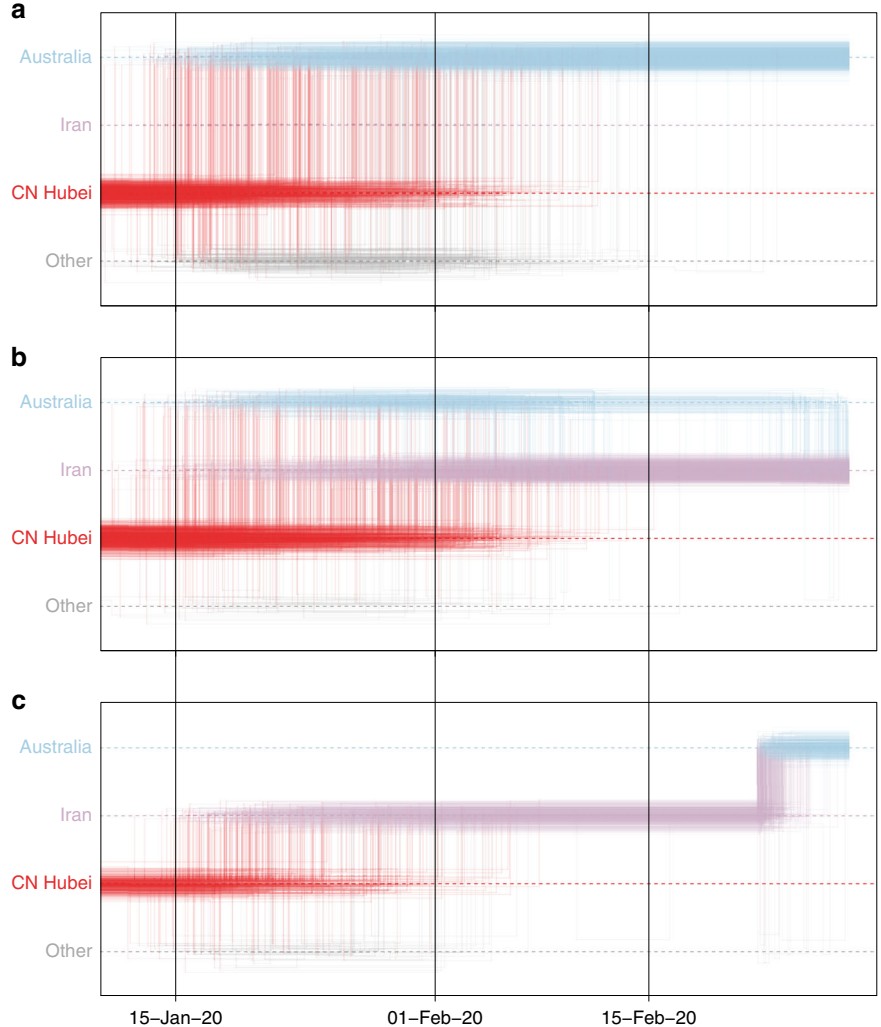

**Fig. 3 Markov jump trajectory plot depicting the ancestral transition history between locations from Hubei up the sampling location for an Australian genome (EPI_ISL_412975) in lineage B.4.** The reconstructions use **a** sampling location only, **b** travel origin location, and **c** sampling location and travel history. The trajectories are summarized from a posterior tree distribution with Markov jump history annotation in the same way as in Fig. 2.

analyses (Fig. 6 and Supplementary Fig. S7). While introductions from Hubei represent the dominant pattern when using sampling location only (Fig. 6a), this is far more pronounced for the other analyses that include travel history. Using sampling location suggests unrealistic dispersal from locations, such as Australia and The Netherlands, that largely disappear when using travel history data, without or with unsampled diversity (Fig. 6b, c). In the travel-aware analyses, European countries experience more introductions from Italy, as well as more directly from Hubei. As also illustrated by the specific examples (Figs. 2 and 3), the considerable number of secondary transmissions from Italy and Iran is revealed by using travel history data (Fig. 6b).

The substantial addition of unsampled taxa from these relatively undersampled locations does not further contribute to this pattern (Fig. 6c), but adding unsampled diversity from Hubei does remove an unexpected dispersal from The Netherlands to Taiwan (Fig. 6b, c).

**Retrospective genome availability fills specific sampling gaps but still benefits from incorporating travel history.** We assembled the 282 genome data set based on genomes available in GISAID on March 10. Because many sequences with sampling times before this date have become available retrospectively,

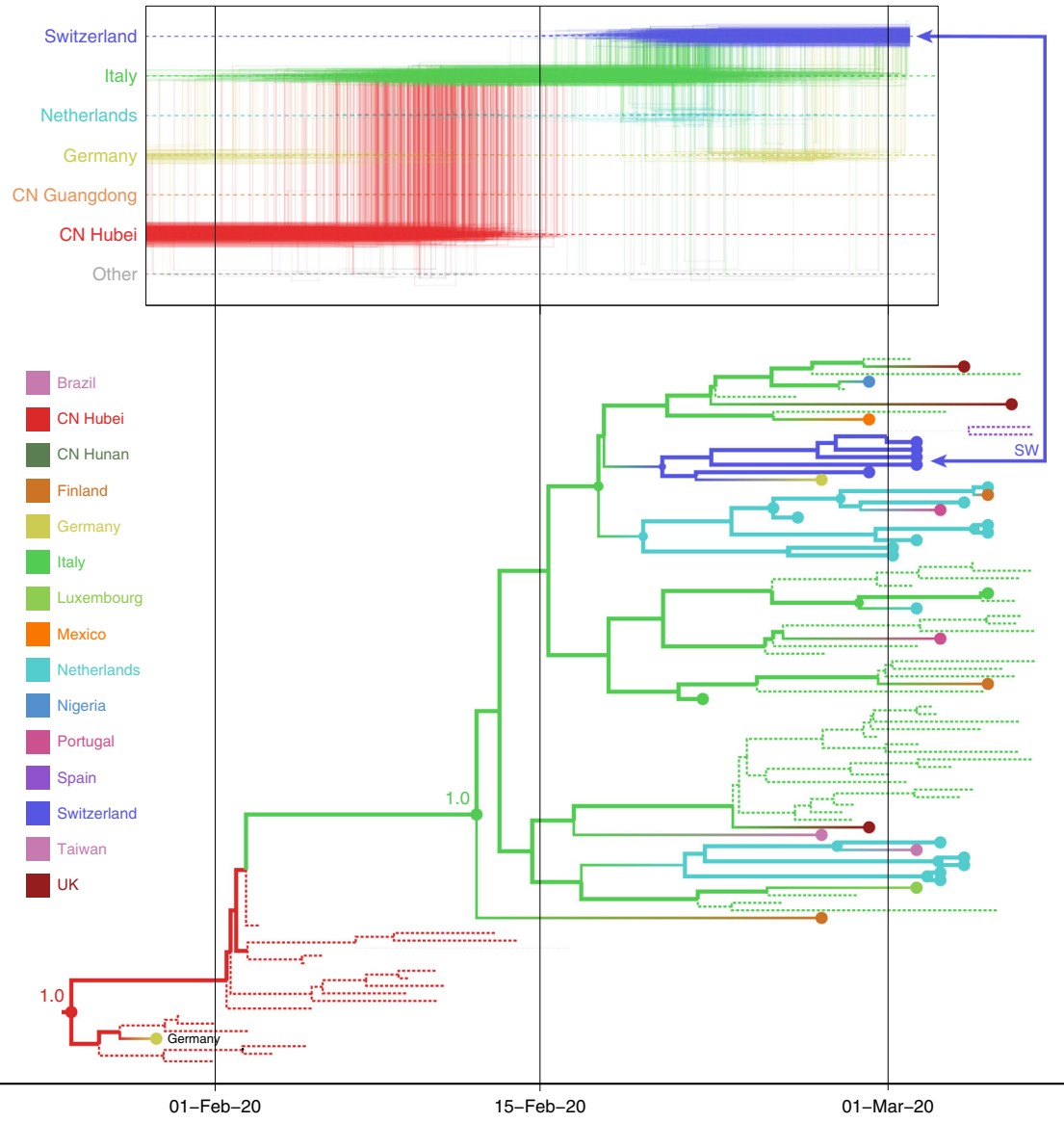

**Fig. 4 Markov jump trajectory plot as in Fig. 3 for a Swiss genome (EPI_ISL_413021) in lineage B1, and B1 subtree for the Bayesian phylogeographic analysis incorporating travel data and unsampled diversity.** Dotted lines represent branches associated with unsampled taxa assigned to Italy and Hubei, China. The tip for the Swiss genome corresponding to the trajectory is indicated with an arrow. The basal German virus is labeled. The value at the root and the common ancestor of the Italian clade represents the posterior location state probability.

we also compiled a larger data set by downsampling the available genome sequences with a sampling time up to March 10 ~4 months after this date. We focus on 43 of the 44 locations represented in the 282 genome data set, but include Shanghai instead of Fujian (cfr. "Methods"). The downsampling procedure mitigates sampling bias for many locations but not all (Supplementary Fig. S5c), and we therefore incorporate travel history for the same subset of genomes as in the 282 genome data set. The travel-aware analysis of the 500 genome data set (Fig. 6d) results in highly consistent overall dispersal dynamics compared to the previous travel-aware reconstructions without or with unsampled diversity (Fig. 6b, c), e.g., in terms of the seeding patterns from Hubei, and the considerable number of secondary transmissions from Italy and Iran. Because of the larger sampling in this data set, a limited number of additional dispersal events are inferred. The presence of 16 additional Italian viruses in the European clade (Supplementary Fig. S8) of the 500 genome data set confirms the Italian ancestry of the clade in much the same way as

the unsampled Italian taxa in the 282 genome analysis (Fig. 4). Incorporating travel history information remains important to establish the Iranian ancestry of the cluster, in which viruses were sampled from travelers returning to Australia, New Zealand, and Canada (Supplementary Fig. S9). Interestingly, the single genome from Iran included in the 500 genome data set is part of this cluster (Supplementary Fig. S9), as predicted by the 282 genome analysis with unsampled taxa (Fig. 5).

## Discussion
International travelers had a central role in the early global spread of the SARS-CoV-2 virus. To track whether COVID-19 cases were new imports or community transmission, detailed travel histories were collected from many of the early patients. To date, however, phylogeographic approaches using discrete trait reconstruction have not been able to fully incorporate travel history data. Researchers had to select whether to assign a sample to the location of sampling, typically the home country, or to the

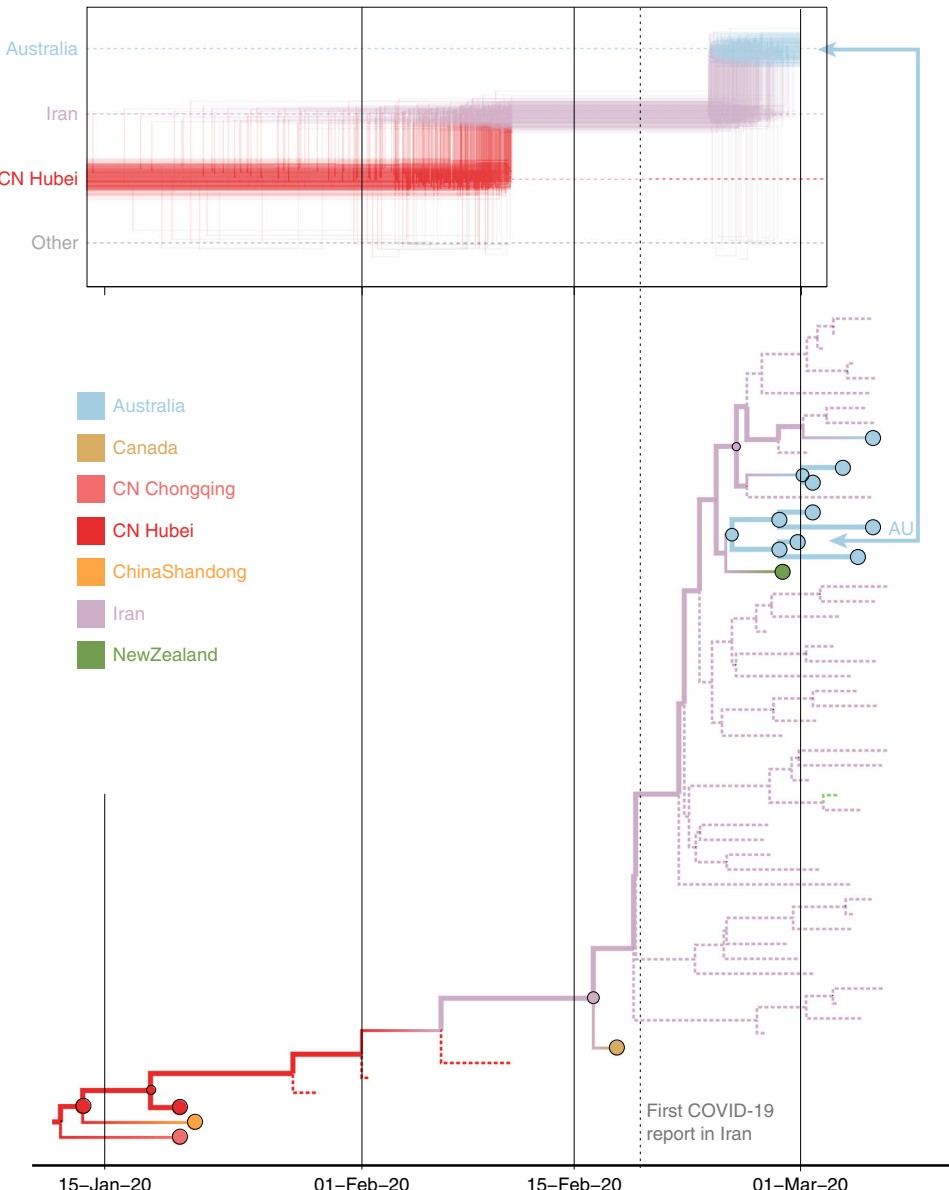

**Fig. 5 Markov jump trajectory plot as in fig. 3 for the Australian genome (EPI_ISL_412975) in lineage B.4 and B4 subtree for the Bayesian phylogeographic analysis incorporating travel data and unsampled diversity.** Dotted branches in the phylogeny are associated with unsampled taxa assigned to Iran and Hubei, China. The tip for the Australian genome corresponding to the trajectory is indicated with an arrow. The vertical dotted line represents the first report of COVID-19 in Iran.

location visited by the traveler. Either way, half of the information was lost. During a period when there were major gaps in the availability of SARS-CoV-2 genomes from many key locations, losing half of the spatial information provided by travelers has been suboptimal. Here, by developing a phylogeographic approach that introduces ancestral nodes in the phylogeny that are associated with locations visited by travelers, we provide a method to formally recapture all the rich information provided by travelers. Most importantly, we demonstrate that the travel-aware approach can dramatically improve phylogeographic inferences about the specific country-to-country paths followed by the SARS-CoV-2 virus during the early stages of global spread. As expected, the inclusion of travel history data is most informative when travelers are arriving from locations, such as Italy and Iran that experienced early SARS-CoV-2 outbreaks, but for which few genome sequences were available, resulting in large gaps in the phylogeny. In addition to illustrating the benefit of including

travel information in such cases, we also demonstrate that the travel-aware reconstructions are associated with a higher posterior predictive accuracy.

We retrieved travel history data for ~20% of the early available genomes, but many more travel-based introductions may be undocumented. While such information is sometimes available in GISAID records, this is not commonly included metadata and there is otherwise no specific resource available to retrieve such information. This may at least partly be explained by concerns about the risk of patient identification. Travel history data may be particularly important when analyzing low diversity data using Bayesian joint inference of sequence and traits because sharing the same location state can contribute to the phylogenetic clustering of taxa. In general, it is crucial to consider the uncertainty of Bayesian time-measured phylogenetic reconstructions because, even for sparse sequence information, a single-tree sample (such as the MCC tree) will appear highly resolved with branching

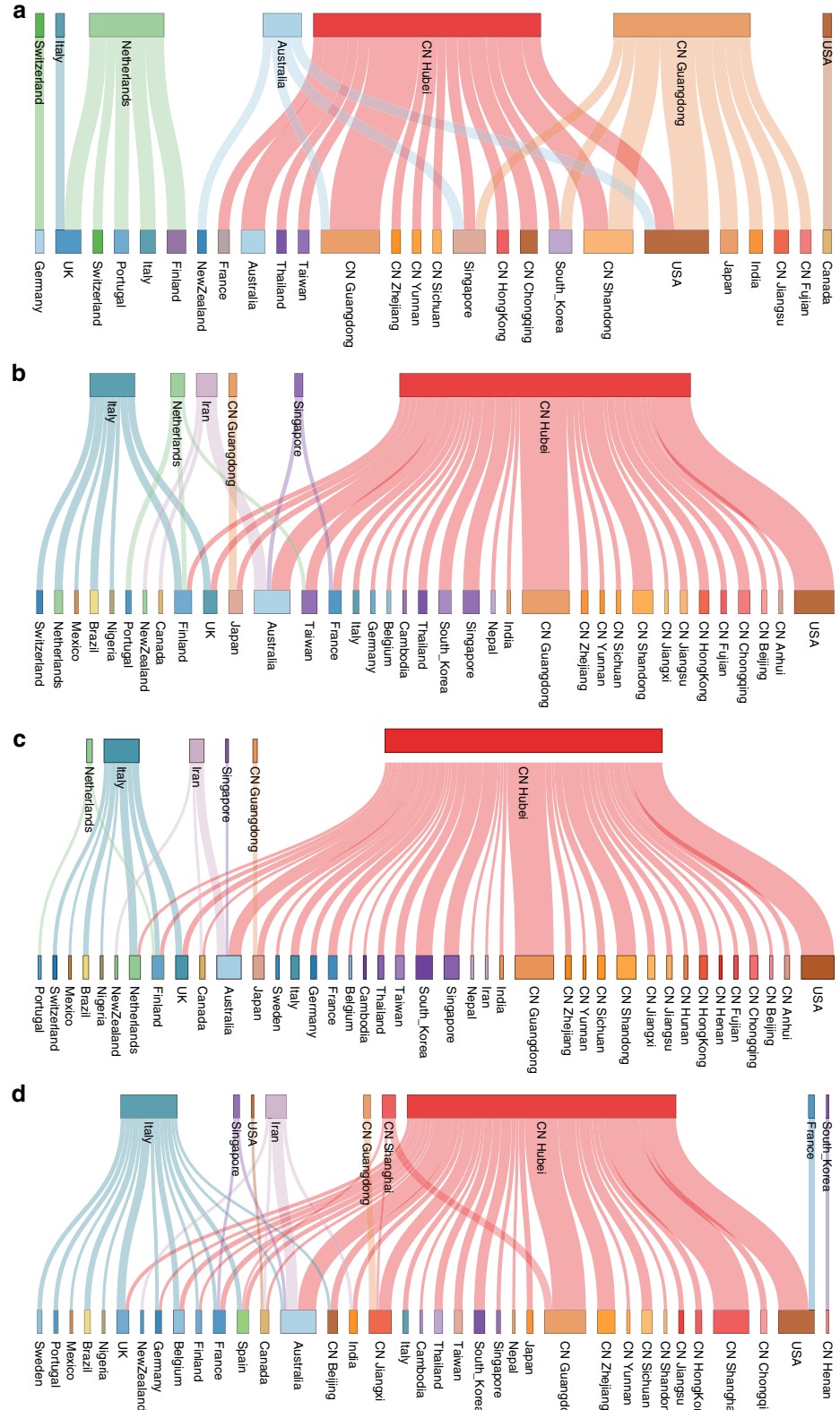

**Fig. 6 Sankey plots summarizing Markov jump estimates for the analyses of the 282 genome data set.** The reconstructions use **a** sampling location only, **b** sampling location and travel history, and **c** sampling location and travel history with unsampled diversity, and **d** for the analysis of the 500 genome data set using sampling location and travel history. The plots show the relative number of transitions between origin (top) and destination (bottom) locations. We note that locations may both be origin locations (in the top row) and destination locations (in the bottom row), and there is no temporal order for the transitions involved. For summaries that show all transitions to and from a location connected to that particular location, we refer to the circular migration plots in Supplementary Fig. S7.

structures that may not be supported by substitutions (Fig. 1). However, this is only a single transmission hypothesis compatible with the data, while many other hypotheses will be plausible as should be reflected by the different topologies in the posterior and hence by low node support values. For this reason, we resorted to posterior summaries that focus on the location-transition patterns in the ancestral history of single taxa or on Markov jump estimates, both averaging over all plausible trees.

We intentionally examined the performance of our methods on an early SARS-CoV-2 data set that was heavily burdened by spatiotemporal sampling bias. However, as we are studying a pandemic as it unfolds, and as new SARS-CoV-2 sequences continue to be generated globally at an explosive pace, there are constant opportunities to reassess the probability of conclusions drawn from earlier data sets and to further expand on them. We made use of this by compiling a second data set ~4 months after constructing the original data set. The retrospective availability of more genomes from Hubei for example removes to a large extent the need for unsampled diversity from the pandemic origin in phylogeographic reconstructions. This updated data set also allows assessing the validity of specific details of the early reconstructions. For example, the data set includes one of the two Iranian genomes that had become available in GISAID, and this virus clusters with viruses from Australian travelers returning from this country (Supplementary Fig. S9), reinforcing the Iran–Australia connection observed in our travel history reconstructions. The availability of additional data from Italy also supports the spatial connections inferred from early traveler data and from unsampled Italian diversity (Figs. 2 and 4, and Supplementary Fig. S8).

While downsampling genomic data from locations in unbalanced data sets has become a common practice[17,20], we present here an alternative approach that adds unsampled taxa to assess the sensitivity of inferences to sampling bias. We emphasize that even though the inclusion of unsampled taxa is informed by epidemiological data, these unsampled taxa should never be considered as additional observations. The unsampled diversity reveals alternative hypotheses that may not be captured by the available genome sampling, but are worth considering in the context of biased sampling. However, reconstructions using unsampled taxa do not provide evidence for any single hypothesis with the same weight as actual genomic data.

Despite the encouraging results, including the recovery a well-established pathway of migration in another empirical example (Zika virus, Supplementary Text S2), we envision reconstructions built with unsampled taxa as being exploratory in nature, and most useful as added support for conclusions drawn independently from other analytical approaches, for example, evolutionary simulations or epidemiological studies.

It is important for future users of these methods to understand exactly how different kinds of empirical data are used to determine where unsampled taxa will attach to the phylogenetic backbone of sampled genomes. The first important aspect is the relative positioning in time of unsampled tips, which in our case is drawn from distribution curves of estimated prevalent infections over time. So, this together with the relative abundance of unsampled taxa by location is informed by epidemiological data. Second, the locations of unsampled taxa will determine their clustering when jointly inferring the phylogeny based on sequences and discrete location traits. In this respect, unsampled taxa will preferentially cluster with taxa representing the same locations, either sampled genomes or other unsampled ones. However, unsampled taxa can, and do in our experience, branch off lineages representing different location states. The relative preference for which location transitions this involves will be determined by the matrix of transition rates of the discrete trait

continuous-time Markov chain (CTMC), which in our case are informed by covariates, such as air travel. This implies that in the SARS-CoV-2 analysis unsampled taxa can be positioned with taxa from countries that are highly connected by airline travel, given the importance of air travel in the early spread of the virus. Finally, the branch lengths, or time it takes for unsampled taxa to find a common ancestor with other taxa, will be influenced by the coalescent prior. We opted for a simple exponential growth coalescent prior in our analyses, but more flexible tree priors are available that can also be informed by epidemiological data, e.g., ref. [24].

By formally accommodating the possibility of unsampled diversity in our phylogenetic reconstructions, we provide alternative scenarios for how SARS-CoV-2 spread globally and entered specific countries. Most importantly, many early introductions in different locations were likely from Hubei, in line with modeling estimates that point at the underdetection of exported COVID-19 cases from Wuhan[25]. In addition, our findings reinforce estimates shaped by travel history that point at early introductions into both Italy and Iran, two countries that are not well represented by genomic sampling, and subsequent transmission events from these countries to other locations. Due to the low genomic variability of SARS-CoV-2, it may be more appropriate to refer to unsampled transmission chains rather than unsampled diversity because many unsampled taxa may represent highly similar or even identical genomes. With the large number of SARS-CoV-2 genomes now available, the question arises how scalable the incorporation of unsampled taxa will be. For computationally expensive Bayesian inferences, the approach may need to go hand in hand with downsampling procedures or more detailed examination of specific sublineages. The complementarity with downsampling is suggested by the 500 genome data set for which downsampling can mitigate, but not fully remove, sampling bias. In our case however, the incorporation of travel history data corrected for particular remaining large sampling gaps (e.g., from Iran). Furthermore, as averaging over all plausible phylogenetic "placements" of large numbers of "volatile" unsampled taxa can be a challenging task, further developments are needed to make the estimation more efficient for larger-scale data sets.

We firmly acknowledge that many aspects of these analyses require further detailed examination and refinement in other pathogen systems, with different types of data gaps and sampling biases. For example, we used an arbitrary threshold for the ratio of available genomes to case counts, in order to decide which locations required representation by unsampled taxa (without accounting for differences in reporting rates for case counts), so it would be useful to investigate how sensitive the reconstructions are with respect to such decisions. Our spatial diffusion GLM may benefit from 2020 air travel data that are impacted by travel restrictions. The time dependency imposed by travel restrictions could potentially be modeled with an epoch version of the discrete trait CTMC[26]. This could also be important for the asymmetry factor we included for transitions from Hubei, as these will be severely impacted by the travel ban imposed on January 23[27]. Finally, in addition to our posterior predictive accuracy assessment, simulation studies would greatly assist in evaluating the performance of the phylogeographic reconstructions in other controlled scenarios.

In conclusion, we demonstrate how travel history data can be formally integrated into discrete phylogeographic reconstructions and that this, together with accounting for unsampled diversity, can mitigate spatiotemporal sampling bias in reconstructions of the early spread of SARS-CoV-2. More research is needed on the specifications of such analyses, and we hope that this work will stimulate developments to further integrate epidemiological

information and other data sources into phylodynamic reconstructions.

## Methods

**SARS-CoV-2 genome data sets and associated travel history**. To focus on the early stage of COVID-19 spread, we analyzed SARS-CoV-2 genome sequences and metadata available in GISAID on March 10th[8]. We curated a data set of 305 genomes by removing error-prone sequences, keeping only genomes with appropriate metadata, and a single genome from patients with multiple genomes available. We assigned each genome a global lineage designation based on the nomenclature scheme outlined in Rambaut et al.[28] using pangolin v1.1.14 (https://github.com/hCoV-2019/pangolin), lineages data release 2020-05-19 (https://github.com/hCoV-2019/lineages). We aligned the remaining genomes using MAFFT v.7.453[29] and partially trimmed the 5′ and 3′ ends. All sequences were associated with exact sampling dates in their meta-information, except for one genome from Anhui with known month of sampling. Upon visualizing root-to-tip divergence as a function of sampling time, using TempEst v.1.5.3[30] based on an ML tree inferred with IQ-TREE v.2.0-rc1[31], we removed one potential outlier. The root-to-tip plots without the outlier are shown in Supplementary Fig. S3. We formally tested for temporal signal using BETS[32]. The final 282 genomes were sampled from 28 different countries, with Chinese samples originating from 13 provinces, one municipality (Beijing), and one special administrative area (Hong Kong), which we considered as separate locations in our (discrete) phylogeographic analyses. Phylogenetic signal in the data set was explored through likelihood mapping analysis[33] (Supplementary Fig. S4).

We searched for travel history data associated with the genomes in the GISAID records, media reports, and publications and retrieved recent travel locations for 64 genomes (22.5%, Supplementary Table 2): 43 traveled/returned from Hubei (Wuhan), 1 from Beijing, 3 from China without further detail (which we associated with an appropriate ambiguity code in our phylogeographic analysis that represents all sampled Chinese locations), 2 from Singapore, 1 from Southeast Asia (which we also associated with an ambiguity code that represents all sampled Southeast Asian locations), 7 from Italy, and 7 from Iran. In this data set, Italy is better represented by recent travel locations than actual samples ($n = 4$) and Iran is exclusively represented by travelers returning from this country. For 46 out of the 64 genomes, we retrieved the date of travel, which represents the most recent time point at which the ancestral lineage circulated in the travel location.

In order to examine (i) to what extent our reconstructions could be updated by the genome data that has become available retrospectively for the same locations and the same time period before March 10 ~4 months after this date, and (ii) how sampling bias can be mitigated by downsampling from the larger collection of available genomes, we assembled an additional data set of 500 genomes. For this purpose, SARS-CoV-2 genomes were downloaded from GISAID on June 23, 2020 and processed according to the COG-UK pre-analysis pipeline (https://github.com/COG-UK/grapevine). Briefly, sequences were aligned to the reference sequence Wuhan-Hu-1 (Genbank accession number NC_045512) using Minimap2 v.2.17[34]. Problematic sites were masked (https://virological.org/t/issues-with-sars-cov-2-sequencing-data/473), and sequences with <95% coverage or an overabundance of mutations were removed. Due to the availability of a relatively large amount of genomes from Shanghai and its importance in international air travel, we considered genomes for this location instead of Fujian to maintain the same phylogeographic dimensionality (44 locations) as in the 282 genome data set. We included the same 64 genomes with travel history used in the 282 genome data set, and for the remaining 436 genomes, we performed a subsampling relative to time and geographic location from the sequences sampled before Mach 10, 2020. To maximize the temporal signal with minimal geographic bias, the genomes were selected such that the 500 sequences were distributed as evenly as possible in each epi-week for which samples were available. Within each week, sequences were sampled proportionally to the cumulative number of cases for that location on March 10. Despite this sampling procedure, the resulting number of genomes by location as a function of case counts still indicates sampling biases (Supplementary Fig. S5c). The root-to-tip divergence plot and likelihood mapping plot are shown in Supplementary Figs. S3 and S4 respectively.

**Incorporating travel history in Bayesian phylogeographic inference**. Discrete trait phylogeographic inference attempts to reconstruct an ancestral location-transition history along a phylogeny based on the discrete states associated with the sampled sequences. In our Bayesian approach, the phylogeny is treated as random which is critical to accommodate estimation uncertainty, when confronted with sparse sequence information. Here, we aim to augment these location-transition history reconstructions on random trees with travel history information obtained from (returning) travelers. When such information is available, the tip location state for a sequence can either be set to the location of sampling, as is done in the absence of such information, or the location from which the individual traveled (assuming that this was the location from which the infection was acquired). Neither of these options is satisfactory: using the location of sampling ignores important information about the ancestral location of the sequence, whereas using the travel location together with the collection date represents a data mismatch, and ignores the final transitions to the location of sampling. These events are

particularly important when the infected traveler then produces a productive transmission chain in the sampling location.

Incorporating information about ancestral locations cannot be achieved simply through the parameterization of the discrete diffusion model, which follows a CTMC process determined by relative transition rates between all pairs of locations that applies homogeneously (or time inhomogeneously[26]) along the phylogeny. Instead, we need to shape the realization of this process according to the travel histories by augmenting the phylogeny with ancestral nodes that are associated with a location state (but not with a known sequence), and hence enforce that ancestral location at a particular, possibly random point in the past of a lineage. Depending on the time at which the ancestral node lies, it may fall on a terminal branch leading to the tip associated with travel history, or before nodes representing common ancestors with other taxa. Further, the location state associated with the ancestral node can also be ambiguous, allowing equal or weighted probability to be assigned to multiple possible locations[35]. We illustrate this procedure for an empirical example that includes 9 SARS-CoV-2 genomes in Fig. 7.

The empirical example includes two genomes from Hubei, four from Australia, and three from Italy. Travel history is available for five genomes (one sampled in Italy and four in Australia), and Fig. 7a demonstrates how this information is incorporated. When a sampled traveler returned from location $i$ to location $j$, we denote time $T_{i \to j}$ as the time when the traveler started the return journey to $j$. At this time point in the ancestral path of the tip (indicated with arrows for the five relevant tips), we introduce an ancestral node and associate it with location $i$, in order to inform the reconstruction that at this point in time the lineage was in location $i$. The upper arrow represents the information introduced for the traveler that returned from Hubei to Italy. The same procedure is applied to the four genomes from travelers returning to Australian from Hubei, Iran, Southeast Asia, and Hubei again (from top to bottom in Fig. 7a).

In subsequent panels, we compare a travel-aware reconstruction (Fig. 7b) to a reconstruction using the standard sampling location (Fig. 7c), and a reconstruction using the location of origin for the travelers (Fig. 7d). Using the location of sampling (Fig. 7c) results in an unrealistic Australian ancestry and two transitions from Australia to Italy, likely because Australia is represented by the largest number of tips. Using the location of travel origin, Fig. 7d results in a reconstruction that better matches the travel-aware reconstruction in terms of inferring an ancestry in Hubei, but misses transitions along four tip branches and differs from the reconstruction including travel history for the upper two Italian genomes. Specifically, it implies a transition from Hubei for the Italian patient that does not have travel history.

We note that $T_{i \to j}$ can be treated as a random variable in case the time of traveling to the sampling location is not known (with sufficient precision). We make use of this ability for the genomes associated with a travel location, but without a clear travel time. In our Bayesian inference, we specify normal prior distributions over $T_{i \to j}$ informed by an estimate of time of infection and truncated to be positive (back-in-time) relative to sampling date. Specifically, we use a mean of 10 days before sampling based on a mean incubation time of 5 days[36], and a constant ascertainment period of 5 days between symptom onset and testing[37], and a standard deviation of 3 days to incorporate the uncertainty on the incubation time. Finally, we indicate that not only information about the sampled traveler can be incorporated, but also about prior transmission history. We apply this for two cases in our data set. One of the genomes was sampled from a German patient, who was infected after contact with someone who came from Shanghai. The person traveling from Shanghai was assumed to be infected after being visited by her parents from Wuhan a few days before she left. In this case, we incorporate Wuhan (Hubei) as an ancestral location with an associated time that accounts for the travel time from Shanghai with a number of additional days and associated uncertainty. Another genome was obtained from a French person, who had been in contact with a person who is believed to have contracted the virus at a conference in Singapore[38]. In this case, we incorporate Singapore as an ancestral location with a known travel time (Supplementary Table 3).

**Incorporating unsampled diversity in Bayesian phylogeographic inference**. To investigate how unsampled diversity may impact phylogeographic reconstructions, we include in our Bayesian inference of the 282 genome data set taxa that are associated with a location, but not with observed sequence data. We identify undersampled locations by considering the ratio of available genomes to the cumulative number of cases for each location (obtained from Our World in Data, https://ourworldindata.org/coronavirus-source-data). To keep all available data, we opt not to downsample genomes, but to add a number of unsampled taxa to specific locations in order to achieve a minimal ratio of taxa (sampled and unsampled) to cumulative number of cases. In Supplementary Fig. S5a, we plot the number of available genomes against the number of cases on March 10th, 2020 on a log–log scale. In our case, we set the minimal ratio arbitrarily to 0.005. Although higher ratios may be preferred, this comes at the expense of adding larger numbers of unsampled taxa, and hence computationally more expensive Bayesian analyses. Our choice for the minimal ratio requires adding 458 taxa for 14 locations (colored symbols in Supplementary Fig. S5b), so ~1.6 times the number of available genomes. Most of the unsampled taxa are assigned to Hubei ($n = 307$), followed by Italy ($n = 47$), Iran ($n = 40$), and South Korea ($n = 30$). For comparison, we also

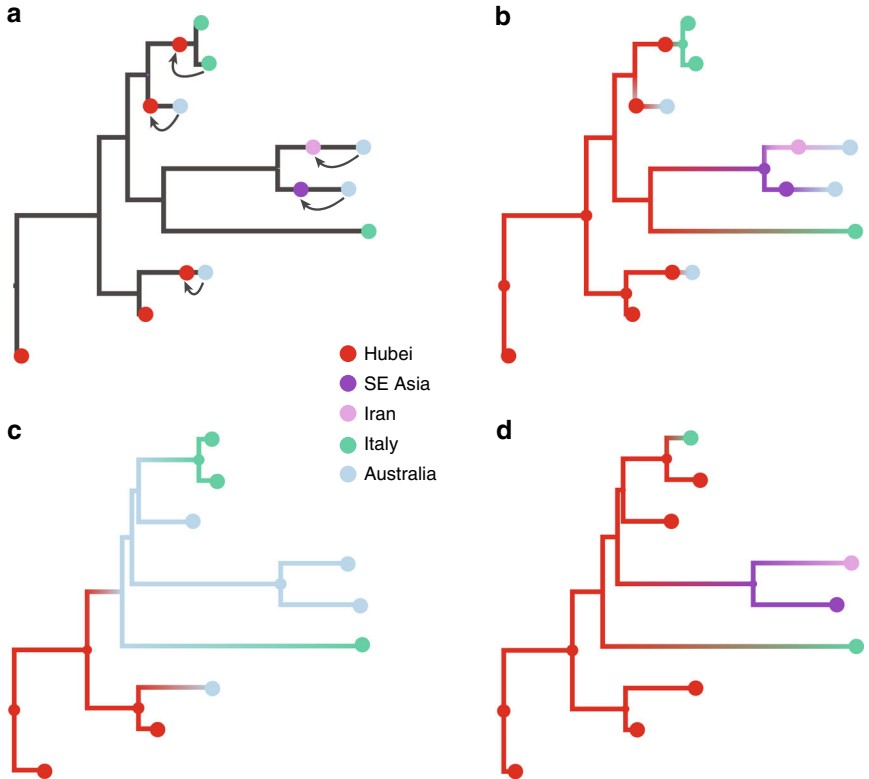

**Fig. 7 Incorporating travel history data in phylogeographic reconstruction. a** The concept of introducing ancestral nodes associated with locations from which travelers returned. The ancestral nodes are indicated by arrows for five cases relating them to the genomes sampled from the travelers. The ancestral nodes are introduced at different times in the ancestral path of each sampled genome. **b–d** The results from analyses using sampling location and travel history, sampling location only, and travel origin location, respectively. The branch color reflects the modal state estimate at the child node. There is some topological variability, but only involving nodes that are poorly supported.

plot the number of available genomes against the number of cases for the 500 genome data set in Supplementary Fig. S5c.

We integrate over all possible phylogenetic placements of such taxa, using standard Markov chain Monte Carlo (MCMC) transition kernels. In the absence of sequence data, time of sampling represents an important source of information for the analysis in addition to sampling location. Here, we use epidemiological data in order to estimate a probabilistic distribution for the sampling times of unsampled taxa. Specifically, we follow Grubaugh et al.[5] in estimating the number of prevalent infectious individuals on day $t$ ($P_t$), by multiplying the number of incident infections up to day $t$ by the probability that an individual who became infectious on day $i$ was still infectious on day $t$:

$$P_t = \sum_{i=1}^{t-1} I_i(1 - \gamma(t-i)) + I_t, \qquad (1)$$

Where $\gamma(t-i)$ is the cumulative distribution function of the infectious period. We also follow Grubaugh et al.[5] in modeling the infectious period as a gamma distribution with mean 7 days and standard deviation 4.5 days. Based on the estimated distributions of prevalent infections for the relevant locations over the time period of our analysis, we specify exponential or (truncated) normal prior distributions on the sampling times of unsampled taxa (Supplementary Fig. S6), and integrate over all possible times using MCMC in the full Bayesian analysis. Using a different, small empirical example, we illustrate the concept of including unsampled diversity in phylogeographic reconstruction and how it contributes to uncovering viral migration pathways (Supplementary Text S2).

**Bayesian phylogeographic inference incorporating global mobility.** We implement our approach to incorporate travel history in discrete phylogeographic inference in the BEAST framework (v.1.10.4[39]). In this framework, we assume that discrete trait data **X**, in our case location data associated with both sampled and unsampled taxa, and aligned molecular sequence data **Y** arise according to CTMC processes on a random phylogeny **F** with the following model posterior distribution:

$$\Pr(\mathbf{F},\mathbf{\Lambda},\mathbf{T},\phi|\mathbf{X},\mathbf{Y}) \propto \Pr(\mathbf{X}|\mathbf{F},\mathbf{T},\mathbf{\Lambda})\Pr(\mathbf{Y}|\mathbf{F},\phi)$$
$$\Pr(\mathbf{F})\Pr(\mathbf{T})\Pr(\mathbf{\Lambda})\Pr(\phi), \qquad (2)$$

where $\Pr(\mathbf{X}|\mathbf{F},\mathbf{T},\mathbf{\Lambda})$ and $\Pr(\mathbf{Y}|\mathbf{F},\phi)$ represent the discrete trait likelihood and sequence likelihood, respectively, **T** is the additional time information for the

ancestral nodes, **Λ** and $\phi$ characterize the discrete trait and molecular sequence CTMC parameterizations along **F**, respectively.

Likelihoods $\Pr(\mathbf{X}|\mathbf{F},\mathbf{T},\mathbf{\Lambda})$ and $\Pr(\mathbf{Y}|\mathbf{F},\phi)$ are calculated using Felsenstein's pruning algorithm[40], a computation that is efficiently performed using the high-performance BEAGLE library[41]. We note that for the travel histories, the ancestral locations and times are introduced only for evaluating the discrete trait location likelihood $\Pr(\mathbf{X}|\mathbf{F},\mathbf{T},\mathbf{\Lambda})$. The ancestral locations and times do not affect the sequence likelihood $\Pr(\mathbf{Y}|\mathbf{F},\phi)$, nor the likelihood of the coalescent model we use as our tree prior $\Pr(\mathbf{F})$.

For the sequence data, we use the HKY nucleotide substitution CTMC model[42], with a proportion of invariant sites and gamma-distributed rate variation among sites[43], a strict molecular clock model, and an exponential growth coalescent tree prior. The uncertainty in collection date for 1 genome was accommodated in the inference by integrating their age over the respective month of sampling. Our discrete location diffusion model involves 44 locations, represented by a limited number of sampled (and unsampled) taxa and ancestral nodes associated with travel locations. In order to avoid having to estimate a huge number of location-transition parameters in a high-dimensional CTMC, and to further inform the phylogenetic placement of unsampled taxa, we adopt a GLM formulation of the discrete trait CTMC that parametrizes the transition rates as a function of a number of potential covariates[17]. As covariates, we consider (i) air travel data, (ii) geographic distance, and (iii) an estimable asymmetry coefficient for Hubei to account for the fact that the early stage of COVID-19 spread was dominated by importations from Hubei (with underdetected cases of COVID-19 probably having spread in most locations around the world[25]). The air travel data consist of average daily symmetric fluxes between the 44 locations in January and February, 2013 (International Air Transport Association, http://www.iata.org). The geographic distance covariate only considers distances for pairs of locations in the same continent, which are based on centroid coordinates. We estimate the effect size of each of these covariates, as well as their inclusion probability (specifying a 0.5 prior inclusion probability for each covariate).

We approximate the posterior distribution of our full probabilistic model using MCMC sampling. We run sufficiently long chains to ensure adequate effective sample sizes for continuous parameters as diagnosed, using Tracer v.1.7.1[44]. We summarize posterior tree distributions using MCC trees, and visualize them using FigTree v.1.4.4. However, due to the limitations of single-tree representations when facing extensive phylogenetic uncertainty, we also propose new summaries below.

A tutorial explaining how to perform travel-aware phylogeographic analyses in BEAST can be found at http://beast.community/travel_history.

**Posterior predictive accuracy assessment**. We validate the approach of incorporating travel history data through a posterior predictive accuracy assessment. Specifically, we perform a tenfold cross validation that, in each fold, holds out a 10% random partition of the travel history information (the ancestral travel location for tip sequences with travel history data) and estimates the known, but not included, location at their respective times in the past ($T_{i \to j}$, generally the travel return times). Across folds, we measure the prediction accuracy for the withheld ancestral travel locations (i) when including the remaining 90% of the travel history, and (ii) when excluding all travel history data, using the original BS for multistate predictions[45], defined as follows:

$$BS = \frac{1}{N} \sum_{i=1}^{N} \sum_{j=1}^{K} \left( p_{ij} - x_{ij} \right)^2, \tag{3}$$

where $N$ is the number of ancestral location instances we predict in our tenfold validation, $K$ is the number of location states, $p_{ij}$ is the posterior probability for location state $j$ in ancestral location instance $i$, and $x_{ij}$ is the outcome for location state $j$ in ancestral location instance $i$ (1 for the observed location state at the ancestral travel history node and 0 for all other location states). This score represents the mean squared error for the predictions and ranges between 0 for perfect accuracy and 2 for perfect inaccuracy, and is a proper scoring rule that incorporates both discrimination and calibration, arguably the two most important characteristics of prediction[46].

**Phylogeographic visualizations**. Due to the relatively low sequence variability over the short time scale of spread, phylogenetic reconstructions of SARS-CoV-2 are inherently uncertain, which also complicates inferring and interpreting location-transition histories. If nodes in an MCC tree are associated with low posterior support, their conditional modal state annotation will be determined by a limited number of corresponding samples from the posterior tree distribution. The addition of unsampled taxa adds an additional challenge because the absence of sequence data makes them highly volatile in phylogenetic reconstruction, reducing posterior node support to impractically low values for many nodes.

In order to marginalize over phylogenetic clustering in our visualization of phylogeographic history, we generate Markov jump estimates of the transition histories that are averaged over the entire posterior in our Bayesian inference[17,47]. We study the ancestral transition history of specific taxa of interest by summarizing their Markov jump estimates as trajectories over time between a number of relevant states. A new BEAST tree sample tool (TaxaMarkovJumpHistoryAnalyzer available in the BEAST codebase at https://github.com/beast-dev/beast-mcmc) and associated R package constructs these estimates. The BEAST tutorial on incorporating travel history also includes information on how to use these tools (http://beast.community/travel_history). We also visualize posterior expected Markov jumps estimates between all locations using Sankey plots and circular migration flow plots. The latter have been successfully used to visualize migration data[48], including phylogeographic estimates[49]. When summarizing these jumps from analyses that include unsampled diversity, we ignore branches that only have unsampled taxa as descendants. We only plot jumps that have a posterior probability larger than 0.90.

**Reporting summary**. Further information on research design is available in the Nature Research Reporting Summary linked to this article.

## Data availability

BEAST XML input files including the GLM-diffusion model and travel history, with or without unsampled diversity, are available at https://github.com/phylogeography/travelHistory[50]. The SARS-CoV-2 genome data required for running these xmls can be downloaded from www.gisaid.org. Supplementary Tables 2–5 list accession numbers for the genomes used in this study. Case count data were obtained from Our World in Data (https://ourworldindata.org/coronavirus-source-data). Air travel data were obtained from the International Air Transport Association (http://www.iata.org).

## Code availability

The code for running BEAST analyses with travel history is available in the ancestral_path branch of the BEAST codebase available at https://github.com/beast-dev/beast-mcmc. The tree sample tool, TaxaMarkovJumpHistoryAnalyzer, is available from the master branch in the same codebase. The R package to visualize Markov jump trajectories is available at https://github.com/beast-dev/MarkovJumpR[51]. A tutorial explaining how to perform travel-aware phylogeographic analyses in BEAST and how to summarize them can be found at http://beast.community/travel_history.

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

## Acknowledgements

We would like to thank all the authors who have kindly shared genome data on GISAID, and we have included a table (Supplementary Tables 4 and 5) listing the authors and institutes involved. The research leading to these results has received funding from the European Research Council under the European Union's Horizon 2020 research and innovation program (grant agreement no. 725422-ReservoirDOCS) and from the European Union's Horizon 2020 project MOOD (grant agreement no. 874850). The Artic Network receives funding from the Wellcome Trust through project 206298/Z/17/Z. P.L. acknowledges support by the Research Foundation—Flanders ("Fonds voor Wetenschappelijk Onderzoek—Vlaanderen", G066215N, G0D5117N, and G0B9317N). G.B. acknowledges support from the Interne Fondsen KU Leuven/Internal Funds KU Leuven under grant agreement C14/18/094, and the Research Foundation—Flanders ("Fonds voor Wetenschappelijk Onderzoek—Vlaanderen", G0E1420N). M.A.S. and K.G.A. acknowledge support from National Institutes of Health grant U19 AI135995. We also gratefully acknowledge support from NVIDIA Corporation with the donation of parallel computing resources used for this research. This work was supported by the Multinational Influenza Seasonal Mortality Study (MISMS), an on-going international collaborative effort to understand influenza epidemiology and evolution, led by the Fogarty International Center, NIH. The content is solely the responsibility of the authors and does not necessarily represent official views of the National Institutes of Health.

## Author contributions

P.L. and M.A.S designed the study and developed the main methodology. P.L., M.A.S, S.H., V.H., G.B., Á.O., and J.T.M. performed the analysis. C.P. and V.C. contributed data. K.G.A., M.W., M.N., and A.R. advised on the methodology and its application, and contributed to the interpretation of the estimates. All authors discussed the results, edited, and approved the contents of the manuscript.

## Competing interests

The authors declare no competing interests.
