## [Peer Review File · Nature Communications]

Reviewers' Comments:

Reviewer #1:

Remarks to the Author:

The present paper propose a phylogeographic method to incorporate information on where the travelers where at the ancestral nodes by investigating the genome sequences of SARS-CoV-2 samples that were available via GISAID on March 10th. The new method is a very elegant integration of travel history data with genomic information. Based on their rich analysis the authors demonstrate how it is possible to reconstruct accurately the spread based on movement using partial data, and by acknowledging the sampling bias and taking in account of travel information. Obviously, the paper deals with a very timely and relevant topic, and the methodology is brilliant. The only critique is not a technical one on the methods, but on the choice of a dataset that was downloaded from GISAID on March 10th may not be reflective of the pandemic at its early stages, and it would strengthen the paper if the authors would add more genomes since it is possible to compare the reconstruction obtained with this method to how it actually occurred since we have data up to June. It appears reductive and a missed opportunity otherwise. A paper using a similar "not up to date" dataset (although, of course, analyzed with methods that cannot even be compared with the ones proposed here) was highly criticized recently by some of the authors. Moreover, released date of a certain genome on GISAID does not correspond to the collection date of the sample; and this is something that should start to appear clearly stated in papers as some might erroneously think that then the data set corresponds to a snapshot of March 10th.

Introduction

Sentence "In addition, large spatiotemporal biases exist in the available genome data. For instance, about 46 % of currently available genomes have been sampled from the UK whereas Italy, having experienced a similar number of cases and likely an earlier epidemic onset, only represents 0.3% of the genome collection on GISAID." This data needs a ref for GISAID and for the sampling bias (e.g. sampling bias was extensively discussed in PMID: 32412415)

Methods

Please include in supplementary the root to tip divergence plot, as well as the likelihood mapping signal (this analysis is missing and it's important to determine the % of phylogenetic noise in the dataset).

Naturally, reviewing the paper now (end of June) means that the number of sequences available on GISAID has totally changed, perhaps increased 10 or more folds since this paper was submitted. Unfortunately, the dataset of three months ago might not be suitable to definitively understand the phylogenetic relationships. As per PMID32412415, it seems that there was not enough phylogenetic signal in data sets of March 10th to allow correct resolution of the phylogenetic relationships among genomes. It would be very helpful if the authors update this paper with newer data to add more power. Perhaps authors can download all genomes collected up to end of March (seems like genomes so far where collected up to beginning of March). It is arguable the decision of the cutoff to March 10th (which corresponds to more recent genome as of March 4th) as a rationale for that date was not given (beginning of stay-at-home or lockdown, or travel bans? That would make sense). Why then not using March 1st or the 9th? Globally the pandemic is still ongoing unfortunately, and a dataset up to April would still been considered early as many countries were still being seeded back then. It is understandable that some of the genomes might not have enough travel information; so just adding genomes for which that information is available would be acceptable (this would also increase the % for genomes that have travel info overall and perhaps increase precision?). This additional analysis would also be an additional proof to show that the for example the dispersal from locations such as Australia and the Netherlands that disappear using unsampled taxa are indeed unrealistic. It would also be very helpful to see how this framework handles more than 300 genomes since the available

data for SARS-CoV-2 is massive.

Fig 4: it is not clear which clade has posterior support which one not. Please provide in supplementary the same tree with branches colored by the 44 countries and with tips IDS to identify where the genomes are clustering.

Fig 5. please provide more information on how to interpret the plots.

Please provide more information on the new tool provided in the methods: "TaxaMarkovJumpHistoryAnalyzer".

Please diversify colors in figure 7 as the blues and the greens look all the same, and provide a matching figure in supplementary with ID in tips.

Please also provide a link to the xmls to show how the unsampled taxa are added to the analysis.

As many of the tips in the phylogenies reported in figure 7 and 8 seem to be unsampled, please provide phylogenetic and temporal signal information for these phylogenies. How would low signal impact the inference?

"We have also observed that two genomes from Iran are now available in GISAID that cluster with the genomes from Australian travelers returning from this country (data not shown), which reinforces the Iran-Australia connection observed in our travel history reconstructions." It would be helpful to see the data in supplementary. And as far for Italy, there are 148 genomes now in GISAID all collected up to end March (one early April), of course some of these were uploaded in May.

The authors make an interesting observation, that this can be an alternative approach to assess the sensitivity of inferences to sampling bias, and it would be interesting to actually compare this inference with a "down-sampled genomic data from the same locations" since that would be possible and compare the two inferences.

Reviewer #2:

Remarks to the Author:

The authors present several distinct methods for allowing the standard discrete trait phylogeographical method to accommodate features of real-world data sets, thereby improving the accuracy and broadening the applicability of the approach.

While the use of generalized linear models to incorporate informative priors on transition rates between locations exists already in the literature (by the same authors), the remaining two improvements are --- to my knowledge --- completely novel: the explicit inclusion of travel history in the tree, as well as the use of data augmentation to reduce the effect of non-homogeneous sampling. The combination of methods is also novel and interesting. Given the wide use of the discrete phylogeographic method, these advances are likely to be widely appreciated. Additionally, the application to SARS-COV-2 phylogeography is compelling and timely. I am therefore generally happy to recommend the manuscript for publication in Nature Communications.

This said, I do have some minor concerns:

1. The approach of augmenting the data set with unsequenced samples seems like a nicely intuitive

approach to reducing the effects of non-uniform sample collection. However, it is clear that the only way to completely remove this as a source of bias is to include enough additional samples to ensure that the combination of sequenced and unsequenced samples is uniformly distributed across the metapopulation. Instead the authors have chosen to limit the number of simulated samples to ensure a minimum fraction (0.005) of the population is sampled, meaning that the composite sample set is still non-uniform. While I understand that there are practical limitations to incorporating more samples, particularly as they contribute to the computational complexity of the phylogenetic inference, I worry that the results reported may still depend heavily on the specific choice of threshold. I thus recommend that the authors conduct a sensitivity analysis to explore the dependence of their results on this arbitrary threshold.

2. It would be wonderful if the BEAST XML files and other scripts used to analyse this data could be included alongside the manuscript in the interest of reproducibility/transparency. While I understand that the GISAID license agreement prohibits onward dissemination of the sequence data, and that the BEAST files traditionally include sequence data, providing XML files from which the sequences themselves have been redacted would still allow all of the details of the analysis to be made clear and furthermore would make it easy for anybody with access to the GISAID database to repeat the analysis.

Reviewer #3:

Remarks to the Author:

The authors describe a new method for incorporating information about travel history and un-sampled genetic data into phylogenetic inference. The issue of unsampled sequences is well-documented and important. For SARS-CoV2 inference, incorporating travel history information has clear benefits and the manuscript provides a welcome addition to the field. The authors describe this new method and use examples of SAR-CoV2 cases to show that this method provides alternative transmission histories than inferred with methods that do not incorporate travel history or un-sampled data. The authors provide a tutorial online, which is fantastic. The manuscript is well written, and the explanations are clear and Figure 1 provides a very helpful overview of the method.

While I do like these parts of the manuscript, the method is at the moment not sufficiently well tested. At least not based on what is not shown in the paper. Currently, the judgement of whether the approach works, seems to be based on whether reasonable travel histories can be reconstructed from actual data when partly conditioning on the true travel history.

Instead, whether the approach actually works has to be addressed as part of a simulation study, where the truth is known. Additionally, an application to a case with a known travel history, where that information is known from other sources of data (or from omitting some travel history data for the analysis), would be a nice control.

Major:

- The phylogeographic model used, takes the sample numbers as informative and as shown in the manuscript wants, in absence of travel information, to be in a different place in the parameter space. Adding travel information improves migration rate estimates, it is, however, unclear if adding phylogenetic information adds anything to the migration rate estimates purely based on the travel histories. The glm analysis uses only reasonable predictors. To be able to judge whether the method

performs well, the authors should also consider showing that they are able to exclude unreasonable predictors and not just include reasonable ones.

- The authors suggest that travel histories can help to overcome sampling bias. What is sampling bias in this context? Is it not having equal sample numbers or not having equal sampling rates between locations. What happens if the information about travel histories are biased themselves? Additionally, whether travel histories help overcome sampling bias should be addressed using simulations and not from actual data were the truth is not know. That is, simulate forward in time structured SIR (or pure I) trees (and not under neutral trait models), add sampling bias and infer migration rates from the number of migration events only, from the phylogenetic tree only and from both together.

- The authors describe 2 applications of their model, one to resolve the transmission history of a Swiss case and one to resolve the transmission history of an Australian case. These examples are helpful and interesting and provide a useful example of the utility of the model. However, it is unclear whether the authors believe that the new reconstructed histories are true based on other, external data (like contact tracing or other epidemiological information) or whether these new reconstructions merely illustrate new hypotheses. Although the new reconstructions surely seem more logical, this is an important distinction. If these new reconstructions agree with other data about the transmission history, then that information should be added and made more clear. If these new reconstructions should be interpreted more as hypotheses, the authors should soften their language and make that clear.

Minor:

- What is the effect of adding migration events on tree inference? It seems from the text that the inclusion of travel histories essentially conditions on coalescent events involving a particular lineage to happen further in the past than migration events. Does this essentially condition on each individual not having transmitted after completing their travel?

- P3: "These studies have already..." please add examples/references

- P4: "The current sequence diversity..." How is the diversity present limiting our ability to track spread and how is it not just the rate of accumulation?

- P6 "Which we cannot include..." Why not? It seems like it should be possible to include the cruise as a discrete location

- Figure 4: Unclear what the histogram adds and what it really represents

- The information being displayed in Figures 5 and 6 is not immediately apparent. The authors should add more information into the legends of these figures to describe what the horizontal vs. vertical lines mean. I presume that each line represents one Markov jump path, and that vertical lines represent jumps between locations. However, this was not immediately clear, and I think that more explicit descriptions in the legends would be helpful to other readers as well.

Reviewer #4:

Remarks to the Author:

In this paper, Lemey and colleagues describe a new approach for phylogenetic reconstruction analysis that integrates individual travel history data and apply it to the early spread of SARS-CoV-2. They provide specific examples of how inclusion of travel history data in Bayesian phylogeographic inference methods used to reconstruct trees can improve the inferences regarding transmission (and also provide new transmission hypotheses). They also introduce the approach of adding unsampled taxa (instead of downsampling genomic data) to account for geographic sampling bias and rebalance the sampled datasets. They are careful to note that reconstructions using the unsampled taxa are

exploratory and may help to confirm independent findings rather than being assigned the same weight as primary genomic data.

The claims of the paper are novel and of high interest to others studying the genomic epidemiology of emerging viruses such as SARS-CoV-2. Sampling bias is indeed a huge issue given that sequencing is heavily weighted towards specific regions in the world with the laboratories with the capacity to generate huge volumes of sequences (such as the UK), and the low circulating diversity of SARS-CoV-2 lineages this early in the course of a novel emerging pandemic also poses a challenge for phylogenetic reconstructions. The findings of the paper would also likely be applicable to future outbreaks from novel pandemic viruses so also are of interest beyond SARS-CoV-2.

The statistical analyses appear to be robust. Standard software packages were used for the analysis (MAFFT for alignment, IQ-TREE for ML reconstruction, BETS for temporal signal, BEAST for calculation of probabilistic models, etc.). I appreciate that a step-by-step tutorial of how to incorporate travel history data into phylodynamic reconstructions is provided online.

I have a few comments and/or suggestions that will help clarify some of the findings in the manuscript:

1. Given the importance of travel location for these analyses, can the authors clarify the resolution of the sampled locations? For example, were the geographic locations reported in GISAID provided to the district, city, county, state, country level, or did the reported locations vary? If they locations varied, can the authors comment on the impact of spatial geographic resolution on their results?
2. Several assumptions were made in the analysis, such as mean of 10 days before sampling and SD of 3 days as an estimate of time of infection. As the number of sampled genomes were relatively small, was a sensitivity analysis performed to see what impact does varying the mean and/or SD have on the results?
3. The addition of unsampled taxa is an intriguing approach to account for sampling bias and evaluate the sensitivity of phylogenetic inferences. However, as the authors state, common practice is still to downsample genomic data from locations. There is also less risk of downsampling genomic data with SARS-CoV-2 given the low circulating diversity of the virus and the large number of identical or nearly identical regions. It's still unclear whether addition of unsampled taxa improves the "believability" of the reconstructions. I am wondering whether the authors considered directly comparing the 3 approaches: (1) standard phylogenetic reconstruction with inclusion of travel information, (2) downsampling the genomic data to balance the datasets, (3) adding unsampled taxa to balance the datasets. I think that this would be helpful in determination of what method would work best for analysis of early SARS-CoV-2 datasets as well as subsequent more recent datasets that have higher diversity.

Point-by-point response to the reviewers' comments,

Thank you again for submitting your manuscript "Accommodating individual travel history, global mobility, and unsampled diversity in phylogeography: a SARS-CoV-2 case study" to Nature Communications. We have now received reports from four reviewers and, on the basis of their comments, we have decided to invite a revision of your work for further consideration in our journal. Your revision should address all the points raised by our reviewers (see their reports below). In particular, it appears that the concerns can all be addressed with moderate extended analyses and text revision. We encourage you to include more up-to-date COVID-19 genomes in the analysis, if possible.

Answer: We thank the Editor and Associate Editor for providing us with four critical and constructive reviews of our work; each stood quite helpful in its own way. In response to each review, we have significantly revised our manuscript by including more up-to-date COVID-19 genomes and extensive analyses, both to validate the predictive performance of our methods.

Reviewer #1 (Remarks to the Author):

The present paper proposes a phylogeographic method to incorporate information on where the travelers were at the ancestral nodes by investigating the genome sequences of SARS-CoV-2 samples that were available via GISAID on March 10th. The new method is a very elegant integration of travel history data with genomic information. Based on their rich analysis the authors demonstrate how it is possible to reconstruct accurately the spread based on movement using partial data, and by acknowledging the sampling bias and taking in account of travel information. Obviously, the paper deals with a very timely and relevant topic, and the methodology is brilliant. The only critique is not a technical one on the methods, but on the choice of a dataset that was downloaded from GISAID on March 10th may not be reflective of the pandemic at its early stages, and it would strengthen the paper if the authors would add more genomes since it is possible to compare the reconstruction obtained with this method to how it actually occurred since we have data up to June. It appears reductive and a missed opportunity otherwise. A paper using a similar "not up to date" dataset (although, of course, analyzed with methods that cannot even be compared with the ones proposed here) was highly criticized recently by some of the authors. Moreover, release date of a certain genome on GISAID does not correspond to the collection date of the sample; and this is something that should start to appear clearly stated in papers as some might erroneously think that then the data set corresponds to a snapshot of March 10th.

Answer: We thank the Reviewer for the positive evaluation of our methodology and for the useful suggestions. We now include a more up-to-date data set that we discuss in more detail in a reply to a specific comment below about the data set.

Introduction

Sentence “In addition, large spatiotemporal biases exist in the available genome data. For instance, about 46 % of currently available genomes have been sampled from the UK whereas Italy, having experienced a similar number of cases and likely an earlier epidemic onset, only represents 0.3% of the genome collection on GISAID.” This data needs a ref for GISAID and for the sampling bias (e.g. sampling bias was extensively discussed in PMID: 32412415)

Answer: Thank you for suggesting these references. They are now included in these sentences.

Methods

Please include in supplementary the root to tip divergence plot, as well as the likelihood mapping signal (this analysis is missing and it's important to determine the % of phylogenetic noise in the dataset).

Answer: we have now included the likelihood mapping results and the root-to-tip divergence plot for both the original and the new data set in the Supplementary Information. As suggested by the Reviewer's next comment (related to PMID32412415), these data sets have a high percentage of unresolved quartets (about 50%, Supplementary Figure 4), questioning their suitability for standard phylogenetic point estimation. This is in agreement with a high number of poorly supported nodes in the MCC tree that we explicitly demonstrated (Figure 2), and it contributes to the argumentation for using an approach that i) appropriately takes into account the phylogenetic error and ii) integrates additional data sources. The former has motivated us to make an effort in providing summaries that fully accommodate the estimation error. In terms of data integration, it is also important to consider that using both sampling time and location information contributes in shaping our full probabilistic estimate. We further inject information about the spatial dispersal process through suitable covariates in our GLM parameterization.

Naturally, reviewing the paper now (end of June) means that the number of sequences available on GISAID has totally changed, perhaps increased 10 or more folds since this paper was submitted. Unfortunately, the dataset of three months ago might not be suitable to definitively understand the phylogenetic relationships. As per PMID32412415, it seems that there was not enough phylogenetic signal in data sets of March 10th to allow correct resolution of the phylogenetic relationships among genomes. It would be very helpful if the authors update this paper with newer data to add more power. Perhaps authors can download all genomes collected up to end of March (seems like genomes so far where collected up to beginning of March). It is arguable the decision of the cutoff to March 10th (which corresponds to more recent genome as of March 4th) as a rationale for that date was not given (beginning of stay-at-home or lockdown, or travel bans? That would make sense). Why then not using March 1st or the 9th? Globally the pandemic is still ongoing unfortunately, and a dataset up to April would still been considered early as many countries were still being seeded back then. It is understandable that some of the genomes might not have enough travel information; so just adding genomes for which that information is available would be acceptable (this would also increase the % for

genomes that have travel info overall and perhaps increase precision?). This additional analysis would also be an additional proof to show that the for example the dispersal from locations such as Australia and the Netherlands that disappear using unsampled taxa are indeed unrealistic. It would also be very helpful to see how this framework handles more than 300 genomes since the available data for SARS-CoV-2 is massive.

Answer: we appreciate the Reviewer's comment about the data set. The rate at which SARS-CoV-2 genomes have become and are still becoming available implies that any study initiated at a particular point during the pandemic can be updated after its review. March 10th was the date on which our methodology was implemented and ready to be used. In other words, we constructed a data set as soon as we were able to use it.

To address the Reviewer's comment we have compiled a new data set about three months after March 10th. We also used March 10th as the cut-off date for sampling time, for three reasons. First, we wanted to remain consistent with the original data set so that we could use this data set as the down-sampled version in the comparison requested in the Reviewer's final comment (and also requested by Reviewer 4). Indeed, many more genomes with a sampling time prior to this data have now become available retrospectively and the sampling bias in data available during the early stage of the pandemic can to some extent be addressed using down-sampling. For constructing the new data set, we use a systematic down-sampling procedure that maximally attempts to mitigate spatiotemporal sampling bias (presented in the Methods section of the revised manuscript). Second, travel information has frequently been collected for genomes during this early stage of the pandemic, but then generally abandoned during the further course of the pandemic. The travel restrictions that were imposed also imply that there were fewer introductions related to travel. For these reasons, we would not be able to add many genomes associated with travel history after March 10th. Third, even when adhering to this early date, we can -- even with downsampling -- collect a large data set, as requested by the Reviewer. In fact, the new data set includes 500 genomes to demonstrate that the framework indeed is able to handle more than 300 genomes.

We compare the reconstruction on this data set (with the same travel history information) to the original data set with travel history and unsampled diversity and find consistent patterns in terms of the overall dispersal dynamics (Fig. 7, Supplementary Fig 7), as well as specific details (Supplementary Fig. 8&9). For example, the availability of many more Italian genomes in the new 500 genome data set confirms the Italian ancestry of the clade in Fig. 7 that was suggested by the unsampled diversity (Supplementary Fig. 8). Sampling bias can however not be entirely removed by downsampling (Supplementary Fig. 5C) and incorporating travel history diversity remains important in correcting for this, adding further evidence to the usefulness of our approach. The results for the 500 genome data set are reported at the end of the results section because we describe it as an update of available genome data since constructing and analysing our original data set, providing a validation of several aspects of our methodology. We reproduce the updated Figure 7 and its description below for the Reviewer's convenience.

Figure 7: Sankey plots summarizing Markov jump estimates for the analyses of the 282 genome data set (A) using sampling location only, (B) using sampling location and travel history and (C) using sampling location and travel history with unsampled diversity, and (D) for the analysis of the 500 genome data set using sampling location and travel history. The plots show the relative number of transitions between origin (top) and destination (bottom) locations. We note that locations may both be origin locations (in the top row) and destination locations (in the bottom row) and there is no temporal order for these transitions. For summaries that show all transitions to and from a location connected to that location, we refer to the circular migration plots in Supplementary Figure 7.

"Retrospective genome availability fills specific sampling gaps but still benefits from incorporating travel history."

"We assembled the 282 genome data set based on genomes available in GISAID on March 10th. Because many sequences with sampling times before this date have become available retrospectively, we also compiled a data set by downsampling the available genome sequences with a sampling time up to March 10th about four months after this date. We focus on 43 of the 44 locations represented in the 282 genome data set, but include Shanghai instead of Fujian (cfr. Methods). The downsampling procedure mitigates sampling bias for many locations but not all (Supplementary Fig. 5C), and we therefore incorporate travel history for the same subset of genomes as in the 282 genome data set. The travel-aware analysis of the 500 genome data set (Fig. 7D) results in highly consistent overall dispersal dynamics compared to the previous travel-aware reconstructions without or with unsampled diversity (Fig. 7B & C), e.g. in terms of the seeding patterns from Hubei and the considerable number of secondary transmissions from Italy and Iran. Because of the larger sampling in this data set, a limited number of additional dispersal events are inferred. The presence of 16 additional Italian viruses in the European clade (Supplementary Fig. 8) of the 500 genome data set confirms the Italian ancestry of the clade in much the same way as the unsampled Italian taxa in the 282 genome analysis (Fig. 5). Incorporating travel history information remains important to establish the Iranian ancestry of the cluster in which viruses were sampled from travellers returning to Australia, New Zealand and Canada (Supplementary Fig. 9). Interestingly, the single genome from Iran included in the 500 genome data set is part of this cluster (Supplementary Fig. 9), as predicted by the 282 genome analysis with unsampled taxa (Figure 6). "

Fig 4: it is not clear which clade has posterior support which one not. Please provide in supplementary the same tree with branches colored by the 44 countries and with tips IDS to identify where the genomes are clustering.

Answer: It would be highly challenging to make a clear visual that shows all branching patterns with their support values and includes color annotations for 44 locations as well as tip IDs, in particular for the new 500 genome data set. Therefore, we now make the MCC trees available with all annotations on a GitHub repository that accompanies our manuscript (<https://github.com/phylogeography/travelHistory>), so that the trees can be inspected in full

detail by the interested reader. This repo also includes the XML files the Reviewer requests below.

Fig 5. please provide more information on how to interpret the plots.

Answer: We agree that more information about the trajectory plots can be useful. We refer to our reply to a similar comment by Reviewer 3 for more detail on how we addressed this. Briefly, we expanded the information in the caption but also added an explanatory sentence to the Results when first discussing these plots.

Please provide more information on the new tool provided in the methods: “TaxaMarkovJumpHistoryAnalyzer”.

Answer: Assuming that the Reviewer requests more information on how to use this tool to extract Markov jump histories for particular taxa from tree files, we now included this information in our online tutorial on the [beast.community](http://beast.community/travel_history) website (http://beast.community/travel_history). We have also added information on how to use the R package to visualize the Markov jump trajectories.

Please diversify colors in figure 7 as the blues and the greens look all the same, and provide a matching figure in supplementary with ID in tips.

Answer: we had made an effort to use location-consistent coloring in our figures. Because the color scheme uses more similar colors within continents, a subtree that mostly focuses on Europe indeed uses similar blue and green coloring. To address the Reviewer's comment, we now abandon color consistency and use a different color scheme for the subtrees for clarity.

Please also provide a link to the xmls to show how the unsampled taxa are added to the analysis.

Answer: We now include the XML files in the GitHub repository associated with this manuscript. We note that due to GISAID terms of use, we cannot include the actual SARS-CoV-2 sequence data in the XML files. Instead, we include placeholders for the sequences. We also include an XML file for a new analysis of a Zika virus data set we added that specifically illustrates the concept of including unsampled taxa and that is not associated with any data sharing issues.

As many of the tips in the phylogenies reported in figure 7 and 8 seem to be unsampled, please provide phylogenetic and temporal signal information for these phylogenies. How would low signal impact the inference?

Answer: These trees are subtrees of the reconstruction based on the 282 genome data set with unsampled diversity. We now make this more explicit by summarizing the phylogenetic placement of the first virus we focus on in the new Figure 3A. Phylogenetic and temporal signal

can only be obtained for the sampled genome data, and as addressed in a reply to an earlier comment, this is now included in the Supplementary Information for the 282 genome data set (as well as the 500 genome data set). Unsampled taxa can be highly volatile in their phylogenetic placement, which makes it critical for the inference to account for their uncertainty in a Bayesian fashion. For this very reason, we provide inference summaries that represent the uncertainty of the estimates.

“We have also observed that two genomes from Iran are now available in GISAID that cluster with the genomes from Australian travelers returning from this country (data not shown), reinforcing the Iran-Australia connection observed in our travel history reconstructions.” It would be helpful to see the data in supplementary. And as far for Italy, there are 148 genomes now in GISAID all collected up to end March (one early April), of course some of these were uploaded in May.

Answer: the new 500 genome data set includes an Iranian virus as well as several additional Italian viruses. We illustrate how the reconstructions for this new data set are consistent with the hypothesis of the Iran-Australia connection (revealed by including travel history information) and the Italian clade (revealed by including travel history information and unsampled diversity). The relevant section on these results is reproduced above in reply to the first comment on the data set.

The authors make an interesting observation, that this can be an alternative approach to assess the sensitivity of inferences to sampling bias, and it would be interesting to actually compare this inference with a “down-sampled genomic data from the same locations” since that would be possible and compare the two inferences.

Answer: this is an excellent suggestion and using the new data set, we are now able to compare these inferences. It is for this reason that we also use March 10th as the cut-off date for the sampling times of the genomes in the 500 genome data set. We note that some locations remain undersampled, so not all sampling bias can be removed by downsampling. We therefore consider downsampling and incorporating unsampled diversity, as well as including travel history, to be complementary. The relevant section on these results is reproduced above in reply to the first comment on the data set. We also highlight the complementarity of the approaches in the discussion as follows: "The complementarity with downsampling is suggested by the 500 genome data set for which downsampling can mitigate, but not fully remove, sampling bias. In our case however, the incorporation of travel history data corrected for particular remaining large sampling gaps (e.g. from Iran). "

Reviewer #2 (Remarks to the Author):

The authors present several distinct methods for allowing the standard discrete trait phylogeographical method to accommodate features of real-world data sets, thereby improving the accuracy and broadening the applicability of the approach.

While the use of generalized linear models to incorporate informative priors on transition rates between locations exists already in the literature (by the same authors), the remaining two improvements are --- to my knowledge --- completely novel: the explicit inclusion of travel history in the tree, as well as the use of data augmentation to reduce the effect of non-homogeneous sampling. The combination of methods is also novel and interesting. Given the wide use of the discrete phylogeographic method, these advances are likely to be widely appreciated. Additionally, the application to SARS-COV-2 phylogeography is compelling and timely. I am therefore generally happy to recommend the manuscript for publication in Nature Communications.

Answer: we are very grateful to the Reviewer for this encouragement.

This said, I do have some minor concerns:

1. The approach of augmenting the data set with unsequenced samples seems like a nicely intuitive approach to reducing the effects of non-uniform sample collection. However, it is clear that the only way to completely remove this as a source of bias is to include enough additional samples to ensure that the combination of sequenced and unsequenced samples is uniformly distributed across the metapopulation. Instead the authors have chosen to limit the number of simulated samples to ensure a minimum fraction (0.005) of the population is sampled, meaning that the composite sample set is still non-uniform. While I understand that there are practical limitations to incorporating more samples, particularly as they contribute to the computational complexity of the phylogenetic inference, I worry that the results reported may still depend heavily on the specific choice of threshold. I thus recommend that the authors conduct a sensitivity analysis to explore the dependence of their results on this arbitrary threshold.

Answer: we agree that this threshold is arbitrary. We acknowledge this and the fact that it would be useful to examine how sensitive the inference is to such decisions in the discussion. However, based on the various comments by the four Reviewers, we decided to focus most of our efforts for this revision on including an updated downsampled data set and performing a systematic evaluation of the performance of including travel history, both of which further add to the length of our manuscript. In addition, we now include an intuitive empirical example based on a Zika virus data set to demonstrate how including unsampled taxa recovers an established migration pathway, irrespective of a threshold. Specifically, we remove sequences from French Polynesia from this example, which represents the link between Southeast Asia and continental America, and illustrate that even only unsampled taxa restore this connection. We include this

analysis in the Supplementary information to not distract from our SARS-CoV-2 application (Supplementary Text S2).

We realize this may not fully address the Reviewer's comment, but there are limits to what we can do with our SARS-CoV-2 data set. With the currently used threshold for example, we already include many more unsampled ($n = 458$) than sampled taxa ($n = 282$). Practical considerations are likely to make this threshold a specific design choice for each data set/study. So, even if we would be able to find a more optimal ratio, it may be specific for our data set and perhaps not applicable for other data sets of interest. Finally, the fact that our reconstructions using unsampled taxa are now consistent with reconstructions based on an updated downsampled data set provides some reassurance that the threshold we used is reasonable.

2. It would be wonderful if the BEAST XML files and other scripts used to analyse this data could be included alongside the manuscript in the interest of reproducibility/transparency. While I understand that the GISAID license agreement prohibits onward dissemination of the sequence data, and that the BEAST files traditionally include sequence data, providing XML files from which the sequences themselves have been redacted would still allow all of the details of the analysis to be made clear and furthermore would make it easy for anybody with access to the GISAID database to repeat the analysis.

Answer: this is an excellent suggestion. We now include the XML files in a GitHub repository associated with this manuscript (<https://github.com/phylogeography/travelHistory>). We follow the Reviewer's suggestion and include placeholders for the sequences in the XML files. Our tutorial on the beast website (http://beast.community/travel_history) further details how to construct the XML files to include travel history.

Reviewer #3 (Remarks to the Author):

The authors describe a new method for incorporating information about travel history and un-sampled genetic data into phylogenetic inference. The issue of unsampled sequences is well-documented and important. For SARS-CoV2 inference, incorporating travel history information has clear benefits and the manuscript provides a welcome addition to the field. The authors describe this new method and use examples of SAR-CoV2 cases to show that this method provides alternative transmission histories than inferred with methods that do not incorporate travel history or un-sampled data. The authors provide a tutorial online, which is fantastic. The manuscript is well written, and the explanations are clear and Figure 1 provides a very helpful overview of the method.

Answer: we thank the Reviewer for this positive assessment.

While I do like these parts of the manuscript, the method is at the moment not sufficiently well tested. At least not based on what is not shown in the paper. Currently, the judgement of whether the approach works, seems to be based on whether reasonable travel histories can be reconstructed from actual data when partly conditioning on the true travel history.

Instead, whether the approach actually works has to be addressed as part of a simulation study, where the truth is known. Additionally, an application to a case with a known travel history, where that information is known from other sources of data (or from omitting some travel history data for the analysis), would be a nice control.

Answer: the Reviewer raises an important point and we agree that our study would benefit from a formal, systematic evaluation of the method. We carefully thought about how to perform such an evaluation. Although vanilla testing on forward-simulated “data” can be a useful approach for this purpose, it leads to an entirely new set of questions/comments about which scenarios to evaluate and how to parameterize such simulations, and, in the end, most such simulations are poorly informed and lack generalizability to real-world problems. Therefore, we decided to combine the Reviewer’s suggestion (simulation and conditioning on known travel history) in a posterior predictive accuracy assessment. This is a simulation study that is highly informed by the real problem at hand in all aspects except, importantly, in the scientific prediction the study wishes to make. Specifically, we perform a 10-fold cross validation that, in each fold, holds out a 10% random partition of the travel history information (the ancestral travel location for tip sequences with travel history data) and estimates the known, but not included, locations at their respective times in the past (generally the travel return times). Across folds, we measure the prediction accuracy for the withheld ancestral travel locations (a) when including the remaining 90% of the travel history and (b) when excluding all travel history data, using the original Brier score for multistate predictions (Brier 1950). This score represents the mean squared error for the predictions and ranges between 0 for perfect accuracy and 2 for perfect inaccuracy, and is a proper scoring rule that incorporates both discrimination and calibration, arguably the two most important characteristics of prediction (Rufibach 2010). Our evaluation results in a Brier score

of 1.35 without travel history data and 0.70 with 90% of the travel history data. This represents a large, two-fold improvement in discrimination and calibration and clearly demonstrates the advantages of including travel history under conditions where the truth is known. We now include the description of this approach in the methods section and report our findings in the results section.

Major:

- The phylogeographic model used, takes the sample numbers as informative and as shown in the manuscript wants, in absence of travel information, to be in a different place in the parameter space. Adding travel information improves migration rate estimates, it is, however, unclear if adding phylogenetic information adds anything to the migration rate estimates purely based on the travel histories. The glm analysis uses only reasonable predictors. To be able to judge whether the method performs well, the authors should also consider showing that they are able to exclude unreasonable predictors and not just include reasonable ones.

Answer: in the GLM parameterization of the discrete diffusion model, the diffusion rates are a function of the predictors, but it is the phylogenetic information for the viruses from different locations that determines which predictors are included in the model. We have indeed used a sparse set of predictors, but out of the three we consider (air travel, out-of-Hubei flux, and within-continent geographic distances), the within-continent geographic distance predictor is not supported. The posterior inclusion probability of about 0.1 is less than the prior inclusion probability of 0.5, which results in a Bayes factor less than 1 (0.11). Including unreasonable predictors does not serve the purpose of our analysis, but it would only serve the purpose of showing that unreasonable predictors do not get any support. For the purpose of demonstrating this to the Reviewer, we reran the analysis that uses sampling location information and also consider a predictor that represents a randomized version of air travel. This analysis results in the following predictor inclusion probabilities:

- within-continent geographic distances: 0.029
- air travel: >0.999
- out-of-Hubei flux: 0.967
- randomized air travel: 0.114

As expected, there is no support in favor of the randomized air travel predictor. We have not included this in the revised manuscript because various applications of the GLM-diffusion model in previous years, e.g. on human influenza (Lemey et al. 2014), swine influenza (Nelson et al. 2015), Ebola (Dudas et al. 2017), HIV (Hong et al. 2020), but also applications to other traits such as hosts for bat rabies (Faria et al. 2013), have shown that the approach can select a meaningful subset of predictors out of a large number of predictors. We cite the Ebola study as an example of this.

- The authors suggest that travel histories can help to overcome sampling bias. What is sampling bias in this context? Is it not having equal sample numbers or not having equal sampling rates between locations. What happens if the information about travel histories are biased themselves? Additionally, whether travel histories help overcome sampling bias should be addressed using simulations and not from actual data were the truth is not know. That is, simulate forward in time structured SIR (or pure I) trees (and not under neutral trait models), add sampling bias and infer migration rates from the number of migration events only, from the phylogenetic tree only and from both together.

Answer: as addressed in our reply to the first comment by the Reviewer, we have addressed the performance of the travel history approach using a posterior predictive assessment. In this approach, we remove a fraction of the actual travel history data and try to estimate the removed ancestral travel locations. We therefore know what the truth is for the removed locations and we can evaluate whether other travel history shapes the phylogeographic estimate in a way that allows making better predictions of the removed locations. We show that the inclusion of travel history data considerably increases prediction accuracy. In our case, most genomes associated with travel history involve viruses sequenced from travellers from Hubei (Wuhan) and this is also the location that is strongly underrepresented in our data. The fact that the travel history corrects for this specific bias is also illustrated by the circular migration plots in the updated Fig. 7. There could indeed be scenarios where travel history itself contributes to bias. This motivated at least in part our warning in the discussion that many aspects of these analyses require further detailed examination and refinement in other pathogen systems with different types of data gaps and sampling biases.

- The authors describe 2 applications of their model, one to resolve the transmission history of a Swiss case and one to resolve the transmission history of an Australian case. These examples are helpful and interesting and provide a useful example of the utility of the model. However, it is unclear whether the authors believe that the new reconstructed histories are true based on other, external data (like contract tracing or other epidemiological information) or whether these new reconstructions merely illustrate new hypotheses. Although the new reconstructions surely seem more logical, this is an important distinction. If these new reconstructions agree with other data about the transmission history, then that information should be added and made more clear. If these new reconstructions should be interpreted more as hypotheses, the authors should soften their language and make that clear.

Answer: the posterior predictive assessment we refer to in previous replies addresses exactly this issue. It assesses how much better we are doing at recovering the truth, the truth being the withheld locations for a fraction of tips with travel history locations in each fold of our cross-validation. We perform this in a systematic way such that each tip has been removed once as part of the 10% removal in the 10 different runs.

Minor:

- What is the effect of adding migration events on tree inference? It seems from the text that the inclusion of travel histories essentially conditions on coalescent events involving a particular lineage to happen further in the past than migration events. Does this essentially condition on each individual not having transmitted after completing their travel?

Answer: our approach of incorporating travel history does not condition on coalescent events. For a genome that was sampled from a traveller who returned from location X at time Y prior to being sampled, we add an ancestral node at time Y prior to the sampling time of the relevant tip and associate it with location X. Depending on time Y, this ancestral node can fall anywhere in the ancestry of that tip, including also on internal branches (which is the case for a traveller returning from Hubei to Italy at the top of Figure 1 A for example). To emphasize that it is independent of coalescent events we include the following sentence in the description: "Depending on the time at which the ancestral node is introduced, this node may fall on a terminal branch leading to the tip associated with travel history or also before nodes representing common ancestors with other taxa."

- P3: "These studies have already..." please add examples/references

Answer: With 'these' we wanted to refer to the examples with their references in the previous studies. Because this was not clear enough, we now changed this to 'such' studies.

- P4: "The current sequence diversity..." How is the diversity present limiting our ability to track spread and how is it not just the rate of accumulation?

Answer: we are not entirely sure whether the Reviewer refers to the rate of accumulation of the actual number of sequences or the rate of accumulation of the sequence diversity. The current rate of sequence accumulation is certainly not limiting if we consider the > 75,000 genome sequences available in GISAID. Due to a relatively low evolutionary rate however this huge genome sample does not exhibit large diversity -- this is what we refer to as current sequence diversity and the second part of the sentence refers to the relatively low evolutionary rate. The low diversity limits the resolution of genomic reconstructions and we now explicitly add this to the next sentence. The low resolution is illustrated by the relatively high number of nodes with low posterior support. As requested by Reviewer 1, we now also perform likelihood mapping analyses showing that the genomic data is very low in phylogenetic signal.

- P6 "Which we cannot include..." Why not? It seems like it should be possible to include the cruise as a discrete location

Answer: the Reviewer is correct that this could be incorporated as a discrete location, but we cannot define geographic distances or air travel between this location and the other geographic locations, which we require for our GLM parameterization. To avoid having to go into too much

detail about this, we rephrased the sentence as follows: "... keeping only genomes with appropriate metadata and a single genome from patients with multiple genomes available."

- Figure 4: Unclear what the histogram adds and what it really represents

Answer: We apologize for not having made this sufficiently clear. The histogram summarizes the posterior support for all the nodes in the MCC tree. We now clearly state this when describing the reconstruction in Fig. 2. This together with the likelihood mapping requested by Reviewer 1 underscores the need to provide summaries that adequately account for the uncertainty.

-The information being displayed in Figures 5 and 6 is not immediately apparent. The authors should add more information into the legends of these figures to describe what the horizontal vs. vertical lines mean. I presume that each line represents one Markov jump path, and that vertical lines represent jumps between locations. However, this was not immediately clear, and I think that more explicit descriptions in the legends would be helpful to other readers as well.

Answer: We agree that more information about the trajectory plots can be useful. To address this, we have expanded the explanation in the caption of the first figure in which we show such trajectories (Figure 3). In this figure, we also highlight the phylogenetic ancestry for the virus under investigation in the MCC tree cluster; this represents a single jump path from the posterior. Finally, we also add the following sentence to the Results section about the trajectory plots: "the trajectory plots summarize across the posterior distribution the time intervals in the phylogenetic ancestry during which the virus remains in the same location (horizontal lines) and the transitions between two locations (vertical lines)." We reproduce the new Fig. 3 and its caption below for the Reviewer's convenience.

A**B****C****D**
01-Jan-20 15-Jan-20 01-Feb-20 15-Feb-20 01-Mar-20 15

Figure 3: Phylogeographic reconstruction and spatiotemporal ancestry of a virus collected in Switzerland (EPI_ISL_413021). A) Phylogenetic cluster with the Swiss virus shaded in grey in the MCC tree and the same B.1 cluster with branches colored according to posterior modal location states inferred by an analysis using sampling location only. The tip for the Swiss virus corresponding to the trajectory in B is indicated with an arrow. Markov jump trajectory plot depicting the ancestral transition history between locations from Hubei up the sampling location for the Swiss genome using (B) sampling location only, (C) travel origin location and (D) sampling location and travel history. The trajectories are summarized from a posterior tree distribution with Markov jump history annotation. A horizontal line in a trajectory represents the time during which a particular location state is maintained in the temporal-spatial ancestry of the virus; an example of such an ancestry is highlighted in grey in the MCC tree cluster. A vertical line represents a Markov jump between two locations in the trajectory; the most prominent locations in the posterior trajectories are ordered along the Y-axis together with 'Other', which represents all remaining locations. The relative density of lines reflects the posterior uncertainty in location state and transition time between states.

Reviewer #4 (Remarks to the Author):

In this paper, Lemey and colleagues describe a new approach for phylogenetic reconstruction analysis that integrates individual travel history data and apply it to the early spread of SARS-CoV-2. They provide specific examples of how inclusion of travel history data in Bayesian phylogeographic inference methods used to reconstruct trees can improve the inferences regarding transmission (and also provide new transmission hypotheses). They also introduce the approach of adding unsampled taxa (instead of downsampling genomic data) to account for geographic sampling bias and rebalance the sampled datasets. They are careful to note that reconstructions using the unsampled taxa are exploratory and may help to confirm independent findings rather than being assigned the same weight as primary genomic data.

The claims of the paper are novel and of high interest to others studying the genomic epidemiology of emerging viruses such as SARS-CoV-2. Sampling bias is indeed a huge issue given that sequencing is heavily weighted towards specific regions in the world with the laboratories with the capacity to generate huge volumes of sequences (such as the UK), and the low circulating diversity of SARS-CoV-2 lineages this early in the course of a novel emerging pandemic also poses a challenge for phylogenetic reconstructions. The findings of the paper would also likely be applicable to future outbreaks from novel pandemic viruses so also are of interest beyond SARS-CoV-2.

The statistical analyses appear to be robust. Standard software packages were used for the analysis (MAFFT for alignment, IQ-TREE for ML reconstruction, BETS for temporal signal, BEAST for calculation of probabilistic models, etc.). I appreciate that a step-by-step tutorial of how to incorporate travel history data into phylodynamic reconstructions is provided online.

Answer: We thank the reviewer for the positive assessment of our work.

I have a few comments and/or suggestions that will help clarify some of the findings in the manuscript:

1. Given the importance of travel location for these analyses, can the authors clarify the resolution of the sampled locations? For example, were the geographic locations reported in GISAID provided to the district, city, county, state, country level, or did the reported locations vary? If they locations varied, can the authors comment on the impact of spatial geographic resolution on their results?

Answer: The geographic resolution for the genomes available in GISAID varies. For the 500 genomes in the new data set for example, the country of sampling is available for each genome, the admin1 level location is available for 90% of the genomes whereas the admin2 level location is available for 40% of the genomes. This excludes an admin2 level analysis for these data,

which would not be practical or meaningful anyway due to the extremely high dimensionality this would impose on the discrete state model. We opted for a spatial partitioning - country level and admin1 level within China -- that maximally uses the genomic data with its geographic annotation while maintaining still a manageable dimensionality in the GLM-diffusion model (44 states). Using the admin1 level for China allows modelling a more realistic epidemiological scenario with Hubei being the center of epidemic spread, hence also allowing us to incorporate an estimable flux out of Hubei in our GLM-diffusion model. Using admin1 level for other countries would not be meaningful as we only have sparse sampling from these locations in the early stage of the pandemic. The only consideration for an alternative partitioning of our data sets would be to not to use admin1 level information in China, but this would only lead to losing some degree of detail in our reconstructions.

2. Several assumptions were made in the analysis, such as mean of 10 days before sampling and SD of 3 days as an estimate of time of infection. As the number of sampled genomes were relatively small, was a sensitivity analysis performed to see what impact does varying the mean and/or SD have on the results?

Answer: for the genomes associated with travel history, but without a specific travel return time, we indeed use an estimate of time of infection. These estimates are based on the studies that we reference in the manuscript (Lauer et al., 2020 and Ferguson et al., 2020). We understand that concern could be raised when we would fix the time of infection, but we made an effort to make these times random parameters in our model and fully accommodate the uncertainty of the estimates by introducing them as prior probability distributions on the time parameters. We are not aware of studies that contest these estimates, so it remains unclear how one should deviate from these estimates in a sensitivity analysis. In addition, this prior specification is only used for 20 out of 64 genomes associated with travel history.

3. The addition of unsampled taxa is an intriguing approach to account for sampling bias and evaluate the sensitivity of phylogenetic inferences. However, as the authors state, common practice is still to downsample genomic data from locations. There is also less risk of downsampling genomic data with SARS-CoV-2 given the low circulating diversity of the virus and the large number of identical or nearly identical regions. It's still unclear whether addition of unsampled taxa improves the "believability" of the reconstructions. I am wondering whether the authors considered directly comparing the 3 approaches: (1) standard phylogenetic reconstruction with inclusion of travel information, (2) downsampling the genomic data to balance the datasets, (3) adding unsampled taxa to balance the datasets. I think that this would be helpful in determination of what method would work best for analysis of early SARS-CoV-2 datasets as well as subsequent more recent datasets that have higher diversity.

Answer: we thank the reviewer for this excellent suggestion. The 282 genome data set is not suitable for downsampling because locations for which reasonable sequence numbers are included such as Hubei are actually undersampled. Therefore, motivated by a request by

Reviewer 1, we now include an updated data set of 500 genomes with sampling dates up to March 10th. Indeed, many sequences have become available retrospectively since we compiled our original data set, allowing us now to mitigate sampling bias for many locations by downsampling. Interestingly, we are able to demonstrate that the reconstructions for this data set with travel history are consistent with those of the 282 genome data set with unsampled diversity and travel history (Figure 7, also reproduced above in reply to the relevant comment by Reviewer 1). For locations undersampled in the 282 genome data set like Italy, the unsampled taxa or the newly available genomes have a similar impact (Supplementary Figure 8), while for other locations that remain undersampled like Iran, the travel history information helps correcting for this bias in both the 282 and 500 genome data set (Supplementary Figure 9). This is why we did not have to add unsampled taxa anymore to downsampled data set. We believe that downsampling and incorporating unsampled diversity can be complementary approaches, and we now point this out more explicitly in the discussion.

Additional changes

- We reran all our analyses using a strict molecular clock model instead of the previously used relaxed molecular clock model because there was no meaningful Bayes factor support in favor of a relaxed clock and date estimates appeared to be somewhat too recent for specific scenarios under a relaxed clock. We also retrieved exact sampling dates for 17 of the 18 genomes for which we previously only had sampling month available (17 genomes from Shandong, China).
- The circular migration plots were removed in the main text in favor of Sankey plots summarizing the migration dynamics as they can be more easily interpreted. The circular migration plots have been moved to the Supplementary Information (Supplementary Figure 7).
- Because we included additional analyses at the request of the Reviewers in what was an already lengthy manuscript, we moved two figures from the Methods section to Supplementary Information (Supplementary Figure 5 and 6).
- Code and data availability sections have been added.

References

- Brier, Glenn W. 1950. "Verification of Forecasts Expressed in Terms of Probability." *Monthly Weather Review* 78 (1): 1–3.
- Dudas, G., L. M. Carvalho, T. Bedford, A. J. Tatem, G. Baele, N. R. Faria, D. J. Park, et al. 2017. "Virus Genomes Reveal Factors That Spread and Sustained the Ebola Epidemic." *Nature* 544 (7650): 309–15.
- Faria, N. R., M. A. Suchard, A. Rambaut, D. G. Streicker, and P. Lemey. 2013. "Simultaneously Reconstructing Viral Cross-Species Transmission History and Identifying the Underlying Constraints." *Philosophical Transactions of the Royal Society of London. Series B*,

- Biological Sciences* 368 (1614): 20120196.
- Hong, Samuel L., Simon Dellicour, Bram Vrancken, Marc A. Suchard, Michael T. Pyne, David R. Hillyard, Philippe Lemey, and Guy Baele. 2020. "In Search of Covariates of HIV-1 Subtype B Spread in the United States-A Cautionary Tale of Large-Scale Bayesian Phylogeography." *Viruses* 12 (2). <https://doi.org/10.3390/v12020182>.
- Lemey, Philippe, Andrew Rambaut, Trevor Bedford, Nuno Faria, Filip Bielejec, Guy Baele, Colin A. Russell, et al. 2014. "Unifying Viral Genetics and Human Transportation Data to Predict the Global Transmission Dynamics of Human Influenza H3N2." *PLoS Pathogens* 10 (2): e1003932.
- Nelson, M. I., C. Viboud, A. L. Vincent, M. R. Culhane, S. E. Detmer, D. E. Wentworth, A. Rambaut, M. A. Suchard, E. C. Holmes, and P. Lemey. 2015. "Global Migration of Influenza A Viruses in Swine." *Nature Communications* 6: 6696.
- Rufibach, Kaspar. 2010. "Use of Brier Score to Assess Binary Predictions." *Journal of Clinical Epidemiology*.

Reviewers' Comments:

Reviewer #1:

Remarks to the Author:

no further comments

Reviewer #2:

Remarks to the Author:

I'm satisfied with the authors response to both of my comments, and am completely happy for the manuscript to be published in its present form.

Reviewer #3:

Remarks to the Author:

The authors were very responsive to reviewer comments, which is much appreciated. The addition of the posterior predictive check is a nice addition to further validate the method, and the additional information in the figure legends greatly clarified what was being displayed. However, one larger concern remains that while the ancestral state reconstruction is less biased, the migration rates still might be. The easiest way to address this concern would be to include sample numbers as a predictor in the GLM model and show that without the travel histories, the sample numbers are a predictor, but not when performing the analysis with travel histories. In addition, there remain 3 minor points, listed below.

Minor:

- In the analysis, the ancestral state inference is biased when not including travel histories. How can all the glm analyses infer the same predictors if in one case the inferred migration histories are biased?
- Can you add a comment on what a Brier Score of 1.35 in absence of including travel histories means for people applying the migration model without travel histories?
- figure labels 3 ; -> ,

Reviewer #4:

Remarks to the Author:

In this revised manuscript by Lemey and colleagues, the authors address the detail the comments raised by me and other reviewers. In particular, I appreciate that they updated the dataset of genomes and compared addition of unsampled taxa with downsampling.

Reviewer #3 (Remarks to the Author):

The authors were very responsive to reviewer comments, which is much appreciated. The addition of the posterior predictive check is a nice addition to further validate the method, and the additional information in the figure legends greatly clarified what was being displayed. However, one larger concern remains that while the ancestral state reconstruction is less biased, the migration rates still might be. The easiest way to address this concern would be to include sample numbers as a predictor in the GLM model and show that without the travel histories, the sample numbers are a predictor, but not when performing the analysis with travel histories. In addition, there remain 3 minor points, listed below.

We thank the Reviewer for this interesting suggestion. We have now fitted a GLM-diffusion model with a sample size predictor for both origin and destination locations and reproduce the estimated inclusion probabilities and effect sizes below for the Reviewer's convenience.

Table 1: Inferred generalized linear model (GLM) of discrete location transitions when using sampling location only and sampling location augment with travel history. We report posterior inclusion probabilities and posterior mean (95% highest probability density intervals) log conditional effect sizes for air travel, geographic distance, asymmetry out of Hubei, and origin & destination sample size.

Predictor	Sampling location only		Sampling location and travel history	
	Inclusion probability	log conditional effect size	Inclusion probability	log conditional effect size
Geographic distance	0.168	0.059 (-0.033,0.159)	0.087	0.049 (-0.040,0.135)
Air Travel	0.995	0.760 (0.364,1.138)	0.985	0.486 (0.234,0.731)
Asymmetry out of Hubei	0.646	0.541 (-0.043,1.116)	>.999	2.154 (1.466,2.648)
Origin sample size	>.999	1.465 (0.933,2.188)	0.222	0.174 (-0.134,0.523)
Destination sample size	>.999	0.766 (0.560,0.975)	>.999	0.527 (0.527,0.865)

These estimates indicate a considerably reduced contribution of sample sizes to the transitions rates in travel-aware inferences. The sample size at origin locations reduces from maximum posterior inclusion probability (>.999) to a low posterior inclusion probability (0.222). The origin predictor is important for the reconstructions as it shapes the ancestral state estimates in clades (e.g. The Netherlands in the B.1 cluster for sampling location only versus Italy for sampling location and travel history, Figure 3 in the main manuscript). The destination sample size predictor remains associated with a maximum posterior inclusion probability, but has a lower log conditional effect size in the travel aware inference. This is expected because a lot of transitions need to be accommodated to locations represented by many samples (e.g. to genomes from the Netherlands in the B.1 cluster when Italy is the dominant ancestral state when using sampling location and travel history). It is interesting to also note that the support for a flux out of Hubei and its log conditional effect size are only moderate when using sampling location only together sample size predictors. Thanks to the many genomes associated with a travel history from Hubei, the flux yields a high support and a very high effect size irrespective of the inclusion of sample sizes in the travel-aware inference.

We now include these results in the Supplementary information and refer to them in the results section on the GLM parameter estimates as follows: "In Supplementary Text S3, we

also report estimates for the same predictor set expanded with sample sizes as an origin and destination predictor for transitions. This indicates that sample sizes contribute considerably less in travel-aware analyses relative to using sampling location only, suggesting that the travel-aware reconstructions will also be more robust to sample size bias.”

Minor:

- In the analysis, the ancestral state inference is biased when not including travel histories. How can all the glm analyses infer the same predictors if in one case the inferred migration histories are biased?

The results appear to indicate that in the absence of sample sizes as predictor (cfr. the analyses above), the signal for air transportation and a flux out of Hubei is sufficiently strong for the predictors to be included, but their effect sizes are affected as we describe in the Results section.

- Can you add a comment on what a Brier Score of 1.35 in absence of including travel histories means for people applying the migration model without travel histories?

We now add that in practical terms, the odds of identifying the correct ancestral state increases 4.5-fold.

- figure labels 3 ; -> ,

We believe the Reviewer refers to the two instances in the caption of Figure 3 where we use a semicolon. In both cases, we have now separated the sentence into two sentences.